# Cargoes move from *cis* to *trans*-Golgi compartments and concentrate in the TGN before exiting

Marinella Pirozzi [1,4✉], Ilenia Agliarulo [1,4], Rosaria Di Martino[1], Vincenzo Manuel Marzullo [1], Micaela Quarto[2], Gabriele Turacchio [1,3], Galina V Beznoussenko[2], Domenico Russo [1✉], Alberto Luini [1✉], Alexander A Mironov [2✉] & Seetharaman Parashuraman [1✉]

## Abstract

The classical models of intra-Golgi transport envision a movement of cargoes from *cis*- to *trans*-Golgi, followed by their sorting at the *trans*-Golgi network (TGN). During this vectorial transport, the cargoes are processed by sequentially acting glycosylation enzymes. A number of studies challenged the vectorial transport model and proposed alternative transport routes bypassing either directional transport or the TGN. We have re-visited intra-Golgi transport using varied cargo synchronization protocols, employing both imaging and biochemical methods. We find that cargoes move vectorially across the Golgi stack and reach the TGN. Cell type-dependent variations in transport kinetics and the limited resolution of fluorescence microscopy could have influenced earlier discrepant interpretations. Further, we find that exit from TGN is a rate-limiting step leading to the accumulation of cargoes there before their monoexponential exit. These findings support an intriguing model of intra-Golgi transport, which involves classical vectorial transport across the Golgi stack followed by a non-maturation-based transport from the stack to the TGN.

**Keywords** Golgi; TGN; Membrane Transport
**Subject Categories** Membranes & Trafficking; Organelles

## Introduction

The Golgi apparatus is the central organelle of the eukaryotic secretory pathway and plays a key role in the processing and sorting of cargoes (proteins and lipids) received from the endoplasmic reticulum (ER). It is composed of stacks of cisternae endowed with distinct enzymatic compositions that collectively determine the polarized architecture of the Golgi (Klumperman, 2011). According to the classical models of intra-Golgi transport, the cargoes arriving at the Golgi from the ER are transported from the *cis*- to the *medial*- and *trans*-Golgi and then to the *trans*-Golgi network (TGN) in about 10 min (Glick and Luini, 2011). During this transport, the cargoes are processed by hundreds of sequential enzymatic reactions specialized for glycosylation as well as for other post-translational modifications, including palmitoylation, phosphorylation, and proteolytic cleavage (Agliarulo and Parashuraman, 2022). This progressive exposure to sequentially acting enzymes is essential to determine a faithful assembly of glycans that happens in a template-independent manner and plays a fundamental role in a variety of cellular functions (Varki, 2017). After being processed at the Golgi apparatus, the cargo proteins are sorted and shipped via membranous carriers to different cellular compartments, including the endosomes and the basolateral or apical plasma membranes (PM) from the TGN (Guo et al, 2014; Stalder and Gershlick, 2020). Most studies based on cargo pulses generated by synchronization protocols have supported the concept of progressive intra-Golgi transport and processing of cargo proteins and lipids (Bonfanti et al, 1998; Mironov et al, 2001; Trucco et al, 2004).

However, later studies raised questions about the vectorial transport model: Patterson and colleagues, by studying the kinetics of cargo transport through and out of the Golgi based on different technical approaches, have challenged the concept of vectorial transport across the Golgi by cisternal maturation (Patterson et al, 2008). Vectorial transport schemes predict that newly arrived cargoes at the Golgi require a period of residence in the Golgi when they are transported from *cis*-Golgi to TGN, sequentially passing through the Golgi compartments where they are processed by sequentially acting glycosylation enzymes before they finally exit the Golgi region. Moreover, cargo exit within the broadly accepted cisternal maturation model, should occur with linear kinetics. Using a specific set of experimental conditions designed to study synchronized transport at steady state (rather than pulsed) conditions, Patterson and colleagues observed that the cargo proteins reach the Golgi and exit without any measurable delay and with monoexponential kinetics (Patterson et al, 2008). These observations are incompatible with the vectorial transport models outlined above. To explain the kinetics they observed, the authors proposed a radically revised model of intra-Golgi transport, called

[1]Institute of Endotypes in Oncology, Metabolism and Immunology "G. Salvatore"-Second Unit (IEOMI-SU), National Research Council of Italy (CNR), Via P. Castellino 111, Napoli, Italy. [2]IFOM ETS, The AIRC Institute of Molecular Oncology, Via Adamello 16, 20139 Milan, Italy. [3]Institute of Translational Pharmacology (IFT), National Research Council, 67100 L'Aquila and Rome, Italy. [4]These authors contributed equally: Marinella Pirozzi, Ilenia Agliarulo. ✉E-mail: m.pirozzi@ieos.cnr.it; d.russo@ieos.cnr.it; a.luini@ieos.cnr.it; alexandre.mironov@ifom.eu; raman.sp@cnr.it

the rapid partitioning model, which explicitly rejects directionality in transport through the Golgi apparatus and proposes that cargoes enter and exit from all parts of the organelle (Patterson et al, 2008). According to this model, when cargoes enter the Golgi apparatus, they segregate from the Golgi residents by rapidly partitioning into distinct lipid-based export domains, from where they are eventually exported. These export domains are present all over the organelle and not restricted to the TGN, as envisaged by other models of intra-Golgi transport (Glick and Luini, 2011). As a result, each Golgi cisterna behaves as an entry/exit compartment operating with monoexponential kinetics (Fig. EV1). To explain the discrepancies between their observations and published observations of vectorial intra-Golgi transport (Mironov et al, 2001; Trucco et al, 2004), Patterson and colleagues pointed out that most previous experiments on intra-Golgi transport were performed using temperature-based synchronization protocols, and that temperature-induced changes in membrane properties can result in an artifactual, i.e., unphysiological, vectorial transport of cargoes from cis- to trans-Golgi, as supported also by computational modeling (Patterson et al, 2008).

Later studies by Tie and colleagues, using a fluorescence-based "super-resolution" method and employing both temperature-dependent and independent synchronization protocols, concluded cargoes do travel from cis- to trans-Golgi but then they leave for their final destination from the trans cisterna and do not reach the TGN, which is until then generally considered the main station for cargo sorting and delivery to the various compartments where these proteins must perform their function (Tie et al, 2016; Tie et al, 2022).

The conclusions from these studies question the previous transport models and raise queries regarding two aspects of the classical transport models: first, vectorial transport of cargoes from cis-Golgi to TGN, when they are processed by sequentially acting glycosylation enzymes before their exit from the Golgi area; and second, the preeminent role of TGN as the cargo sorting station of the Golgi apparatus. By questioning these well-accepted aspects of Golgi transport, these queries have serious implications for our understanding of Golgi organization and function and present a stumbling block in the ongoing integration of morphological and molecular aspects of intra-Golgi transport, for instance, to the glycosylation function of the Golgi apparatus. Glycosylation is a fundamental and abundant post-translational modification that regulates a myriad of cellular functions. The glycosylation enzymes are distributed in asymmetric gradients within the Golgi apparatus, with early-acting enzymes in the cis-Golgi and the late-acting enzymes in the trans-Golgi (Dunphy and Rothman, 1985), corresponding with the vectorial transport of cargoes as envisaged by classical models of intra-Golgi transport, especially the cisternal maturation model. Analysis of glycosylation in intact cells has shown that this asymmetric localization of the enzymes is essential for proper glycosylation of cargoes (Fisher et al, 2019), especially when the glycosylation pathway involves competitive reactions (Pothukuchi et al, 2021), suggesting vectorial transport is likely important for faithful glycosylation of cargoes (Jaiman and Thattai, 2020). Similarly, before being sorted at the TGN into carriers, several cargoes undergo post-translational modifications, like furin-mediated cleavage, sialylation, sulfation of proteoglycans, or assembly of large complexes like von Willebrand factor at the TGN (Bosshart et al, 1994; Dick et al, 2012; Ferraro et al, 2014;

Molloy et al, 1994; Roth et al, 1985). Further, some cargoes also require TGN46 for their exit (Lujan et al, 2024), suggesting the necessity of cargoes to traverse the TGN before their exit. Indeed, recently, the presence of furin cleavage site in the Spike protein of SARS-CoV2 has been shown to promote the increased infectivity of the viral particles and indeed in the emergence of this new strain of the virus (Ord et al, 2020; Shang et al, 2020), suggesting an obligatory transport through the furin-containing TGN compartment. Thus, given the links between the intra-Golgi transport mechanism and cargo processing reactions, especially glycosylation, we decided to carry out a dedicated study aimed at reexamining the transport route through the Golgi apparatus and establishing a solid basis for our understanding of the main morphofunctional aspects of intra-Golgi cargo transport.

We have examined the kinetic features of cargo transport through the Golgi apparatus using multiple approaches, including transport synchronization methods independent of temperature shifts that were not available at the time Patterson and colleagues performed their experiments (Boncompain et al, 2012) and followed the intra-Golgi transport of cargoes using multiple imaging (fluorescence and electron microscopy) and biochemical methods. We report two main observations: first, in all the experimental conditions used, independently of temperature shifts or pulsed vs steady state traffic protocols, cargoes enter the Golgi stack from the cis face, traverse the organelle to exit from the TGN in about 10–15 min in accordance with the directional transport and sequential processing models of transport. Second, after crossing the stack, cargo reaches the TGN, accumulates there, and then leaves the Golgi region by monoexponential kinetics. The transport out of the TGN thus represents a rate-limiting step in transport through the Golgi-TGN system. The results presented support a scheme according to which TGN must be looked at as a distinct "organelle" from the stacked cisternal part of the Golgi apparatus and emphasize that this needs to be taken into account while developing models of intra-Golgi transport.

The discrepant observations described above were probably due to the cell type used and experimental conditions like the amount of cargo that prevented the observation of a delay before the monoexponential exit of cargoes or the reliance only on fluorescence microscopy, whose resolution limits prevented the observation of cargo exit from the TGN. Collectively, the results presented here support a novel transport scheme according to which cargo progresses linearly through the stack and then enters the TGN, a distinct "organelle" with transport rates and mechanisms different from those operating in the stacked cisternal part of the Golgi apparatus.

## Results

### Newly arrived cargoes exit the Golgi apparatus following a delay

We designed a series of experiments to analyze the kinetics of Golgi exit to discriminate between the vectorial transport and rapid partitioning models. The rapid partitioning model predicts that newly arrived cargoes exit the Golgi apparatus without any apparent delay, as would be expected if the cargoes do not have to traverse the Golgi apparatus before exiting it (Fig. EV1A)

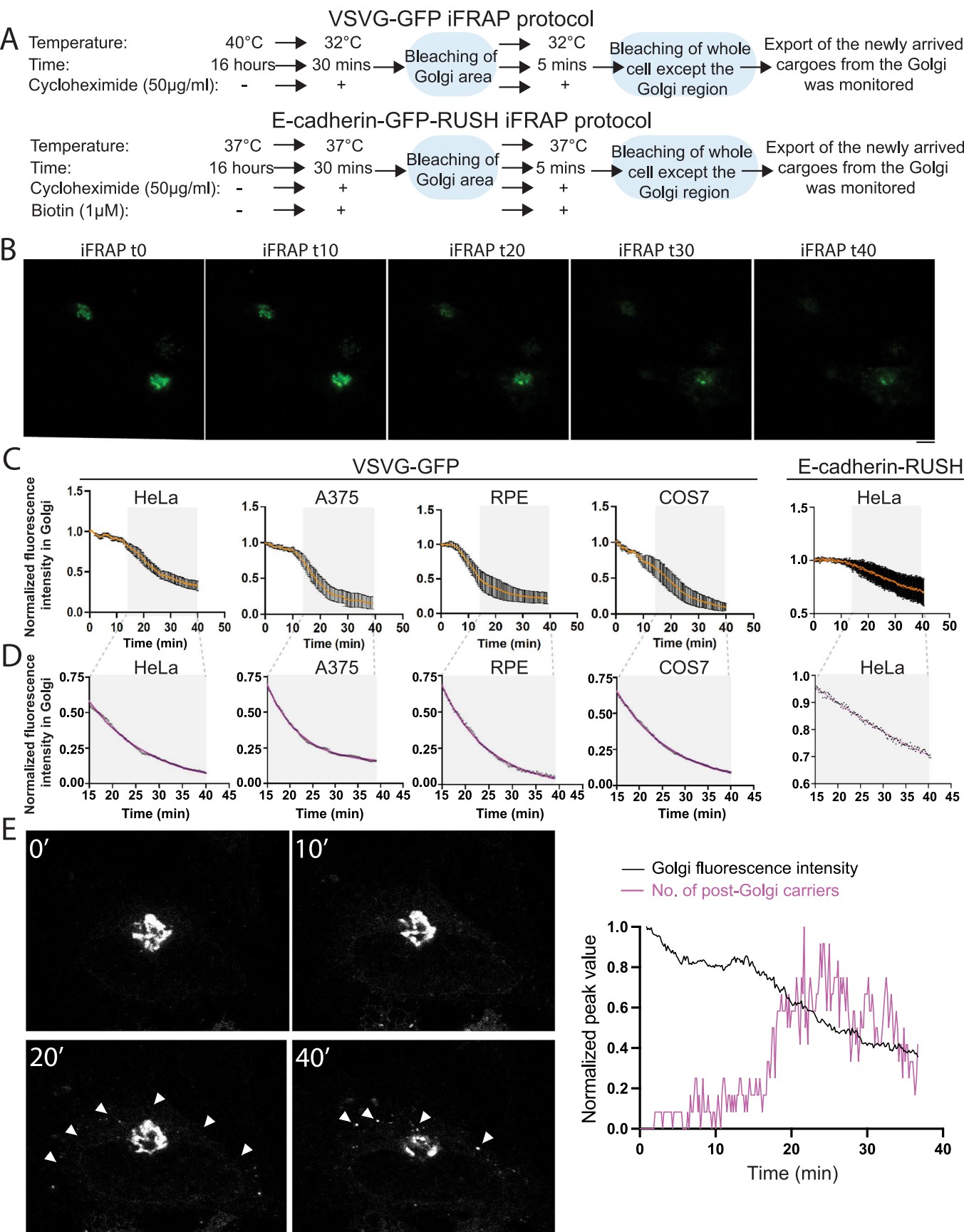

**Figure 1. Newly arrived cargoes exit with a delay from the Golgi.**

(A) Schematic representation of the iFRAP protocol for VSVG or E-cadherin. (B) Cells (the images shown were obtained from HeLa cells) were transfected with VSVG-GFP and the cargo was accumulated in the ER at 40 °C overnight. The cells were then shifted to 32 °C for 30 min to allow the cargoes to enter the Golgi. Then the fluorescence in the Golgi area was photobleached, and the cargo was allowed to enter the Golgi area for 5 min followed by photobleaching of the whole cell fluorescence except the Golgi (iFRAP). Bar: 10 μm. (C) The exit of the newly arrived VSVG from the Golgi area was then monitored and the fluorescence in the Golgi area was calculated over time and represented as normalized to the initial (after iFRAP) fluorescence levels. In parallel, HeLa cells were transfected overnight with E-cadherin-GFP-RUSH constructs. Biotin was then added to facilitate cargo entry into the Golgi, after which the iFRAP protocol (see (A)) was used to track the exit of newly arrived E-cadherin from the Golgi. (mean $+/-$ SE; $n = >3$; see Fig. EV2). The cell lines and the cargo used for the study are indicated. (D) The data from the cargo exit kinetics (C) were extracted to analyze only those between 15 min after iFRAP and 40 min (black). Time frame highlighted by the gray square. The data for the VSVG-GFP and E-cadherin-GFP-RUSH exiting from Golgi fit well with a monoexponential decay (red) with the following rate constants: VSVG in HeLa: 0.077 min$^{-1}$, A375: 0.138 min$^{-1}$, RPE: 0.102 min$^{-1}$, COS7: 0.082 min$^{-1}$; E-cadherin in HeLa: 0.01975 min$^{-1}$. All five data fit had a R-squared value > 0.99. (E) Images from the indicated time points of the cargo exit following iFRAP are shown. Arrowheads indicate post-Golgi carriers. Bar: 10 μm. The adjacent graph compares the exit of cargo from the Golgi (black lines) with the number of carriers present in the cell (pink) over the indicated time period. Carriers represent the fluorescent spots present outside the Golgi area. Source data are available online for this figure.

(Patterson et al, 2008). On the contrary, the vectorial transport models predict a delay before the newly arrived cargoes exit the Golgi (Fig. EV1B). To investigate this, we used several experimental conditions, including precisely those used by Patterson and colleagues in their study. We followed the transport of newly arrived green fluorescent protein (GFP)-tagged temperature-sensitive mutant of vesicular stomatitis virus G protein (VSVG-GFP) in HeLa cells, a cell system that we have used extensively to demonstrate the wave-like transport of cargoes across the Golgi apparatus (Mironov et al, 2001). We expressed VSVG-GFP in HeLa cells and accumulated the protein in the ER by placing the cells at 40 °C for 16 h. The cells were then shifted to the permissive 32 °C for 30 min to allow VSVG-GFP to fill the secretory pathway. The Golgi area was then bleached and fluorescence was allowed to recover for a period of 5 min by the arrival of cargoes from the ER. Then the whole cell, except the Golgi region, was bleached (iFRAP —inverse fluorescence recovery after photobleaching) and the export of the newly arrived cargoes from the Golgi area was monitored by imaging (Fig. 1A,B). The kinetics of the decrease in fluorescence intensity over the Golgi area indicated the presence of two phases—an initial phase when there was little or no reduction in fluorescence, followed by a phase of exponential decay of fluorescence (Figs. 1C,D and EV2A–C). This was similar to what was observed by us using a temperature-dependent cargo transport pulse (Beznoussenko et al, 2023). Next we checked if similar behavior is exhibited by cargoes following a temperature-independent synchronization. To this end we used the now well-established Retention Using Selective Hooks (RUSH) system to synchronize the transport of cargoes (Boncompain et al, 2012). E-cadherin-GFP-RUSH was released to enter the Golgi with the addition of biotin at 37 °C and 30 min later, it was subjected to FRAP followed by iFRAP using protocol described earlier (Fig. 1A). Even in this case, we found that E-Cadherin-GFP-RUSH behaved similar to the VSVG-GFP—there was an initial flat phase where there was no decrease in Golgi-associated fluorescence, and then followed by a monoexponential decay (Fig. 1C,D). While the extent of reduction of Golgi-associated fluorescence was lesser in the case of E-Cadherin-GFP-RUSH, the profile of exit kinetics was similar compared to VSVG-GFP. Thus, we conclude cargoes, independent of their mode of synchronization, stay in the Golgi area for a definite period before exiting.

This initial flat phase of stay observed here contrasts with the observations of Patterson and colleagues (Patterson et al, 2008), but

is consistent with the models that include a vectorial intra-Golgi transport of cargoes (Glick and Luini, 2011). Given the contrasting observations between us and Patterson and colleagues, we studied the same phenomenon in three other cell types—human melanoma cell line A375, retinal pigment epithelial (RPE) cells and also African green monkey kidney fibroblast-like cell line COS7, the cell type used by Patterson and colleagues. Similar curves were obtained in all the cell lines with an initial flat or slow phase followed by a fast exponential decay (Figs. 1C,D and EV2A,B), though the rate of decrease during the flat or slow phase varied among cells and cell lines. While there was almost no reduction in the fluorescence intensity during this phase in A375 and RPE cells, there were variable levels of decrease in HeLa and COS7 cells, with the reduction relatively steeper in COS7 (Fig. EV2A,B). The variable initial phase of fluorescence reduction, especially in COS7 cells (the cell type used by Patterson and colleagues), may have led to the reported apparent absence of the initial flat phase of exit of newly arrived VSVG-GFP from the Golgi apparatus and the conclusion that cargo leaves the Golgi immediately after arrival (Patterson et al, 2008). Potential reasons for this discrepancy are discussed below. We also note that results supporting vectorial trafficking through the Golgi were also reported by Lu and colleagues using conditions different from those used by Patterson and colleagues, thus making comparisons between the results of the two groups difficult (Tie et al, 2016, 2022).

We also examined the formation of cargo-laden carriers from the Golgi as a proxy for the kinetics of exit from the organelle. If the cargoes do exit from the organelle without any delay following their entry, the formation of cargo-containing carriers will peak immediately following the start of the analysis, since the amount of cargo in the Golgi will be maximum at this time point. When high-resolution movies of iFRAP were analyzed, we observed that in the initial time points, few or no cargo-containing structures were moving out of the Golgi (Fig. 1E). After a delay of 10 min, and coinciding with the start of the faster exponential phase of cargo exit, there was an explosive formation of cargo-containing carriers (Movie EV1), and the presence of abundant post-Golgi carriers that transported the cargoes to the PM was seen (Fig. 1E), similar to what has been observed following temperature-based synchronization studies (Polishchuk et al, 2003). The cargo-containing carriers that formed during the faster phase of cargo exit likely correspond to the Golgi to PM transport carriers that were observed and described earlier (Hirschberg et al, 1998; Polishchuk et al, 2003). It

is currently unclear what cargo exit during the initial slow phase may correspond to. It might be that a fraction of cargoes exit directly from the *cis*-Golgi to PM, as was proposed by Patterson and colleagues, or it might correspond to a small fraction of cargoes that reach the TGN by faster kinetics through intercisternal connections (Beznoussenko et al, 2022; Beznoussenko et al, 2014) and exit from the TGN (see below). A further possibility is that this initial reduction in the fluorescence may correspond to the retrograde transport of a fraction of cargo proteins from the Golgi apparatus to the ER that had been reported earlier (Fossati et al, 2014). Indeed, in images where the fluorescence was adjusted post-acquisition to visualize the faint post-Golgi carriers (Fig. 1E), we were able to see a faint recovery of ER-associated fluorescence in the early time points after bleaching, suggesting a minor fraction of cargoes from the Golgi may be retrogradely transported to the ER. In any case, the fraction of cargoes that follow this *slow transport mode* of exit from the Golgi corresponds to a minor fraction of ~10% of the total newly arrived cargoes. The majority of the cargo exits the Golgi in the second phase, in a fast exponential manner. While all the cell lines examined showed this initial slow phase of exit, there was variability in its duration between cell types, and even among samples from the same cell type. For instance, it was close to 5 min in RPE and close to 15 min in HeLa cells (Fig. 1C). The latter is similar to the intra-Golgi transport time (10–15 min) demonstrated by us in HeLa cells following temperature-based synchronized transport (Mironov et al, 2001), suggesting that the initial phase may correspond to the intra-Golgi transport of cargoes. This would imply that the variability is likely due to differences between cell types in the intra-Golgi transport rate, differences in cell states, such as activity levels of signaling pathways (Giannotta et al, 2012), and their subsequent effect on Golgi architecture (Mavillard et al, 2010). Variations in the type and quantity of the cargo being transported may also contribute to it (Beznoussenko et al, 2022). Of note, the observed differences in delay between this study and that of Patterson and colleagues cannot be ascribed to photobleaching-induced artifacts since the protocol used involves the bleaching of the Golgi apparatus-associated fluorescence followed by fluorescence recovery for 5 min (Fig. 1A). The fluorescence recovery is likely due to cargo arriving from ER and shows that the Golgi apparatus is functional. The next photobleaching step (iFRAP) involves the whole cell, except the Golgi, and so is not expected to affect the functionality of the Golgi apparatus. To conclude, cargoes entering the Golgi apparatus are retained in the organelle for a defined period, likely dependent on cell type and cell state, before their exit to the PM.

## Cargoes traverse the Golgi in a vectorial manner

The delay in the exit of newly arrived cargoes is in line with the proposal by several models of intra-Golgi transport that the cargoes have to traverse the Golgi before exiting the organelle. Patterson and colleagues proposed that the observation of directional transport through the Golgi detected in previous experiments (Mironov et al, 2001; Trucco et al, 2004) is likely an artifact resulting from the temperature-induced changes in membrane properties (Patterson et al, 2008). To understand if this was indeed the case, we used the RUSH system to synchronize the transport of cargoes (Boncompain et al, 2012). We used nocodazole-treated cells for this assay since it is known that microtubule depolymerization

by nocodazole leads to the creation of Golgi ministacks that provide a clear distinction between *cis-* and *trans-*Golgi at the fluorescence level (Shima et al, 1997) and thus facilitate the monitoring of the transport of cargoes across the Golgi stacks. The *cis-* and *trans-*Golgi markers we have used in this study (GM130 and TGN46, respectively) were also localized to the *cis-*Golgi and TGN respectively in high-resolution EM studies (Nakamura et al, 1995; Prescott et al, 1997) suggesting that these markers indeed represent two poles of the Golgi apparatus and not segregated zones of a single cisterna. We have used the nocodazole-induced Golgi ministack system successfully to precisely localize proteins within the Golgi apparatus and found that it reflected the data obtained by high-resolution electron microscopy studies (Pothukuchi et al, 2021; Rizzo et al, 2013; Rizzo et al, 2021; Trucco et al, 2004), and also nocodazole-induced ministacks were found to efficiently transport and process the cargoes (Trucco et al, 2004). VSVG-GFP-RUSH or E-Cadherin-GFP-RUSH was retained in the ER and allowed to enter the Golgi ministacks by the addition of biotin (Boncompain et al, 2012). Cells were fixed at different time points after the addition of biotin, and the relative localization of VSVG-GFP-RUSH (Fig. 2A,B) or E-Cadherin-GFP-RUSH (Fig. 2C) with respect to GM130 (a marker of *cis-*Golgi) and TGN46 (a marker of TGN) was used to see if there was a transport of cargo across the stack. As presented in Fig. 2A–C, we could see that with increasing time after the release from the ER, the localization of cargoes moves from the *cis-*Golgi toward the *trans-*Golgi. The cargo that was predominantly present in the *cis-*Golgi at 5 min after the release from the ER was found to be predominantly present in the *trans-*Golgi after 25 min of release from the ER (Fig. 2A–C), which roughly corresponds to the 10–15 min intra-Golgi transport time we observed earlier in the HeLa cells (Fig. 1B). Still there is an apparent delay of 5 min in the intra-Golgi transport time of cargoes as measured by the immunofluorescence assay compared to iFRAP experiments described earlier (Fig. 1C). Whether this is due to the absence of microtubules (following the use of nocodazole) in this assay needs further exploration. Similar results were obtained when the images were analyzed using the published Golgi protein localization by imaging centers of mass (GLIM) methodology (Tie et al, 2022) (Fig. EV2D). Of note, in the later time points, there was substantially more spread of the cargo across the Golgi stack (Fig. 2B,C). This is likely due to the continued arrival of cargo into the Golgi (since once biotin is added, it cannot be washed out due to high-affinity binding with streptavidin), unlike with temperature-based protocols, where this is apparently better controlled. Nevertheless, this study shows that cargoes are indeed transported directionally from the *cis-* to *trans-*Golgi in a temperature-independent fashion.

It is important to note that a recent study has indicated that Golgi ministacks in nocodazole-treated cells are functional after 6 h of treatment, and this can be ascertained by the presence of Giantin in the functional Golgi stacks. While this study does not say that the stacks are "non-functional" after 3 h of treatment, and we and others have used 3 h of nocodazole treatment to restore functional ministacks (Cole et al, 1996, Trucco et al, 2004; Tie et al, 2022). Nevertheless, we also ascertained in the conditions that we use, most of the Golgi ministacks were functional by colocalizing Giantin, GM130 and VSVG in the stacks after 25 min of VSVG trafficking from ER to Golgi. As shown in Fig. EV3A,B, almost all the nocodazole-induced ministacks were positive for all

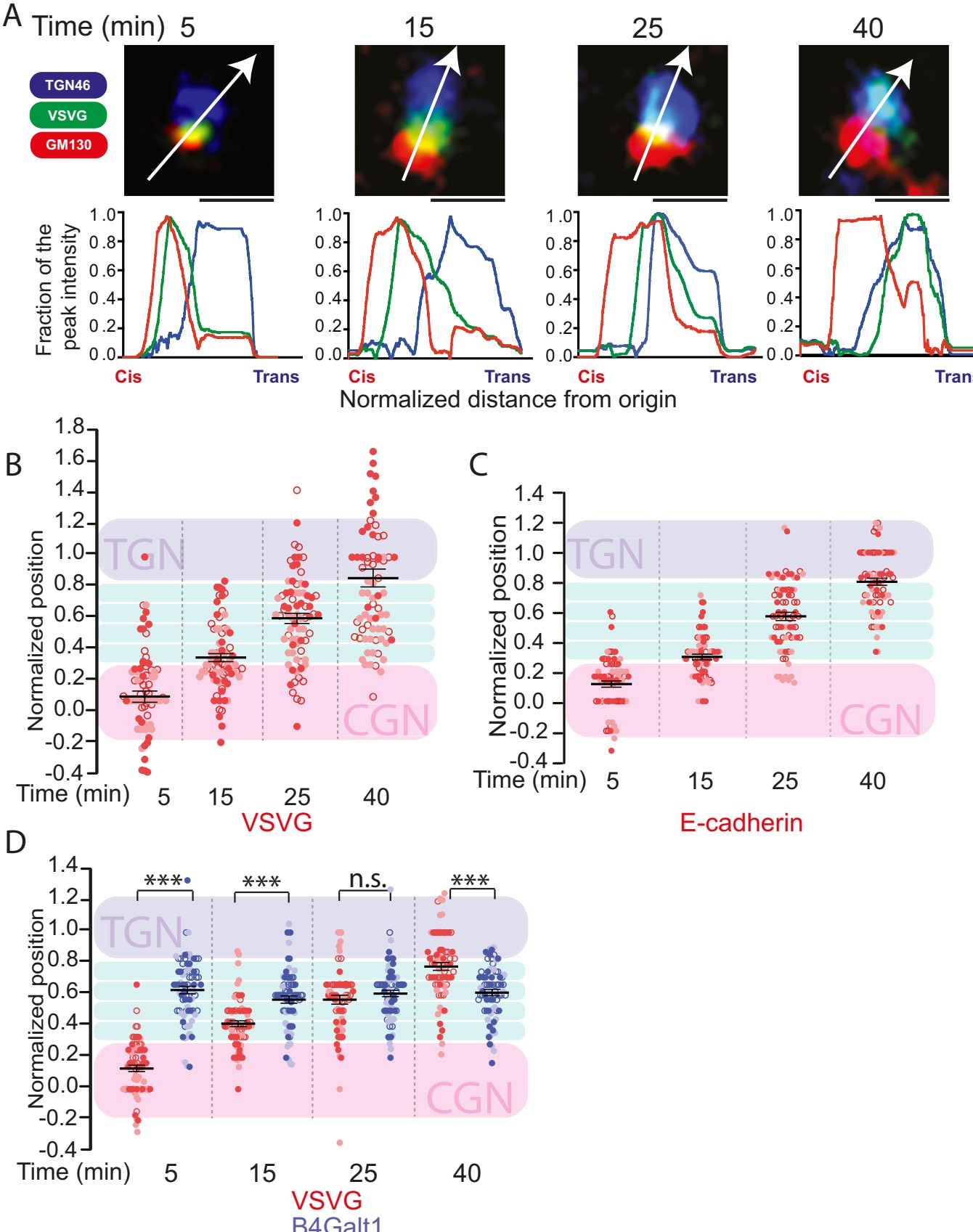

**Figure 2. Cargoes move vectorially from *cis*- to *trans*-Golgi.**

(**A**) HeLa cells were transfected with VSVG-GFP-RUSH constructs overnight and then treated with nocodazole (33 µM) for 3 h at 37 °C. Biotin was then added to cells and at indicated times the cells were fixed and stained for GM130 and TGN46 to mark the *cis*- and *trans*-Golgi, respectively. The relative position of the cargo with respect to GM130 and TGN46 is shown. Bar: 1 µm. (**B**) The position of the cargo and Golgi marker peaks were normalized considering the position of the GM130 peak to be 0 and TGN46 to be 1. The relative positions of cargo peaks over time are plotted (mean $+/-$ SE; Data points ($n > 70$) from three independent experiments are represented by grouped circles in three different shades of the same color). (**C**). HeLa cells were transfected with E-cadherin-GFP-RUSH construct overnight and then treated with nocodazole (33 µM) for 3 h at 37 °C. Biotin was then added to cells and at indicated times the cells were fixed and stained for GM130 and TGN46 to mark the *cis*- and *trans*-Golgi, respectively. The position of the cargo and Golgi marker peaks were normalized considering the position of the GM130 peak to be 0 and TGN46 to be 1. The relative positions of cargo peaks over time are plotted (mean $+/-$ SE; Data points ($n > 70$) from three independent experiments are represented by grouped circles in three different shades of the same color). (**D**) HeLa cells were co-transfected with VSVG-Halotag and GalT-CFP, and the cargo was accumulated in the ER at 40 °C overnight. Cells were treated with nocodazole (33 µM) for 3 h, then shifted to 32 °C to allow the cargoes to enter the Golgi, and fixed at each indicated time. Cells were then stained for GM130 and TGN46 to mark the *cis*- and *trans*-Golgi, respectively. The position of the cargo and Golgi marker peaks were normalized considering the position of GM130 peak to be 0 and TGN46 to be 1. The relative positions of cargo peaks over time are plotted (mean $+/-$ SE; Data points ($n > 70$) from three independent experiments are represented by grouped circles in three different shades of the same color). n.s = not significant (***$P = 1.96e-33$; $3.2e-07$; $6.89161e-07$ [Student's *t* test]). Source data are available online for this figure.

three markers, suggesting that the Golgi stacks that we use are functional.

Further, earlier studies using temperature-dependent protocols had shown that the cargoes exit from the TGN (Griffiths and Simons, 1986; Polishchuk et al, 2003). So, to see if also with temperature-shift independent transport protocols they behave this way, we studied the exit of VSVG-mCherry-RUSH. The cargo was released from the ER by the addition of biotin and the cargo transport was allowed to reach a quasi-steady state by allowing the transport to proceed for 1 h. Then the cells were fixed and stained for *cis*-Golgi (GM130) and *trans*-Golgi/TGN (TGN46) markers. There were tubules containing the cargo VSVG-mCherry-RUSH, that were observed to be leaving the Golgi apparatus like it was observed earlier in temperature-based synchronization studies (Fig. EV3C,D). Visible tubules were rare since the cargoes were not synchronized by accumulating them in the Golgi and we had to score several cells to find one with tubule(s). Moreover, we studied the Golgi ribbon and not the ministacks where the separation between GM130 and TGN46 localizations was not always clearly visible likely due to the convoluted nature of the Golgi ribbon and we only considered cases where the separation between *cis*- and *trans*-Golgi was evident. Regardless, when cargo-containing tubules were observed, they appeared to leave from areas labeled with TGN46 (Fig. EV3C). Note that the resolution of fluorescence imaging is limited (see below) and thus definite evidence of the exit compartment requires a correlative light electron microscopy study. Altogether, these observations show that VSVG (and likely other cargoes) move vectorially from the *cis*- to *trans*-Golgi and exit from the *trans*-Golgi/TGN is in line with the recent observation of the maturation of cargo containing *cis*-Golgi to *trans*-Golgi in yeast cells (Casler et al, 2019; Kurokawa et al, 2019) and also classical observations of the transport of large cargoes across the Golgi (Bonfanti et al, 1998; Weinstock and Leblond, 1974). These classical studies in tissues have shown that procollagen structures and secretory granules become increasingly condensed from cis to *trans*-Golgi and the cargoes present in the secretory vesicles resemble those present in the TGN, suggesting that these vesicles likely originated from there (Weinstock and Leblond, 1974). Further, they are also compatible with the role of ordered Golgi enzyme localization in influencing glycosylation reactions (Fisher et al, 2019; Pothukuchi et al, 2021; Pothukuchi et al, 2019).

Of note, similar observations were also made by Tie and colleagues using fluorescence-based localization methods similar to those described here (Tie et al, 2016; Tie et al, 2022). Both the studies—the one presented here and that of Tie and colleagues—using fluorescence-based methods observed that the peak of the cargo wave did not completely coincide with the TGN peak at least during the time points studied (see below for later time points). This led Tie and colleagues to question the classical models of intra-Golgi transport that proposed an exit from the TGN and they proposed that the cargoes exit directly from the *trans*-Golgi (Tie et al, 2016; Tie et al, 2022). It is to be noted that while we find the mean position of the cargo localization to be in the *trans*-Golgi, there is a spread of cargo from *cis*-Golgi to TGN at later time points (Fig. 2B), suggesting that cargoes do enter the TGN and there is also a likely arrival of cargoes into the Golgi apparatus. The continued presence of cargoes also in the *cis*-Golgi at later time points may lead to an apparent positioning of the peak cargo localization to the *trans*-Golgi. Nevertheless, it does raise the question of whether the cargoes directly exit from the Golgi stack/*trans*-Golgi, which we examined as described below.

## Cargoes exit the Golgi apparatus at the TGN

The classical studies that demonstrated cargo exit from the TGN have mostly used electron microscopy (EM) to study the phenomenon. The recent studies however had used fluorescence-based methods that have a limited resolution. Although Tie and colleagues employed image analysis algorithms that allowed them to overcome this limitation to an extent (Tie et al, 2016; Tie et al, 2022). So, we decided to examine this issue using three different analyses. First, we examined the relative positioning of VSVG in relation to galactosyltransferase, a *trans*-Golgi marker as indicated by Tie and colleagues. (Tie et al, 2022). To this end, HeLa cells were co-transfected with VSVG-Halotag and GalT-CFP and the VSVG-Halotag transport through the Golgi labeled with GalT-CFP at 32 °C was monitored by fluorescence microscopy similar to what was described earlier (Fig. 2D). We observed that VSVG-Halotag initially moved through the Golgi from the *cis*-Golgi towards the TGN. As expected, at early time points, the distribution of VSVG-Halotag differed significantly from that of GalT-CFP. However, as VSVG-Halotag reached the *trans*-Golgi at 25 min, its distribution was not distinguishable from that of GalT-CFP. Surprisingly, prolonged incubation led to a statistically significant separation of the distributions of VSVG-Halotag and GalT-CFP. VSVG-Halotag shifted away from GalT-CFP and towards the TGN46 peak,

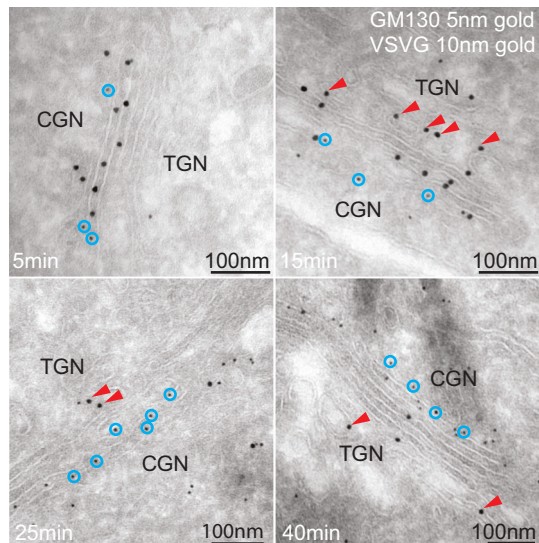

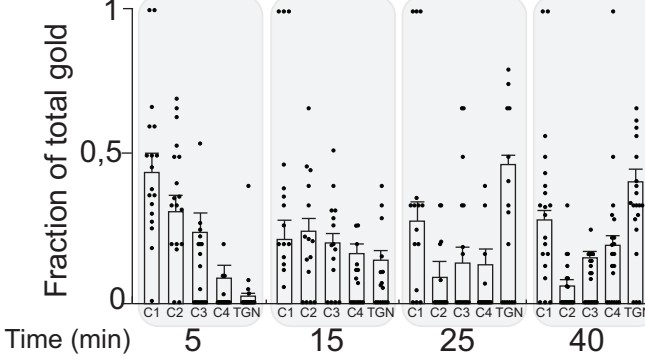

**Figure 3. Cargoes reach the TGN after intra-Golgi transport.**

Upper panel: Cells treated as in Fig. 2A (except for the nocodazole treatment) were fixed and prepared for EM. The presence of VSVG-GFP was visualized by anti-GFP antibody pointed with red arrowsheads (10 nm gold particles) and GM130 to indicate *cis*-Golgi with anti-GM130 antibodies circled in blue (5 nm gold particles). Lower panel: The distribution of VSVG-GFP gold particles across the Golgi stack was quantitated and presented as a percentage of gold particles per cisterna (see "Methods") (mean $+/-$ SE; Data points $n > 15$ stacks). Source data are available online for this figure.

suggesting that VSVG-Halotag has moved out of the *trans*-Golgi toward the TGN (Fig. 2D).

Next, to overcome the resolution limits of fluorescence microscopy and to monitor the transport of VSVG across the Golgi apparatus, we resorted to high-resolution immuno-EM. The transport was synchronized as described earlier (Fig. 2A,B), and the distribution of VSVG-GFP-RUSH across the Golgi apparatus was analyzed by immuno-EM (Fig. 3). This analysis showed that as expected VSVG-GFP-RUSH was mostly present in the *cis*-Golgi (C1 cisterna) 5 min after exiting from the ER. At 15 min however the distribution of the cargo was more spread with all the cisterna containing more or less equal amounts of cargo and a significant fraction of the cargo was also present in the tubulo-vesicular elements present adjacent to the *trans*-Golgi, which correspond to the TGN. Later with increasing time (25 min and 40 min after exit from ER) the total gold particles associated with the Golgi reduced

suggesting an exit out of the Golgi apparatus with a peak accumulation in the TGN likely corresponding to the cargoes moving from the Golgi stack to the TGN. Of note, at these latter time points there was also a peak of cargo concentration in the C1 cisternae of the Golgi likely due to the continued arrival of cargo from the ER. This may have confounded fluorescence studies described earlier (Fig. 2A,B) and those by Tie and colleagues shifting the cargo peak towards *trans*-Golgi rather than TGN. Indeed, at the later time points when cargo arrival is almost nil a presence of cargo peak in TGN is discernible (see below). Thus, high-resolution EM studies indicate that cargoes reach the TGN during their transport across the Golgi.

Next to validate this observation by another means we studied the transport of human growth hormone (GFP-FM4-hGH), which includes four FM domains that tetramerize in the absence of the ligand DD solubilizer and is retained in the ER (Gordon et al, 2010). Upon addition of the ligand, the polymers dissociate and the protein is then transported to the Golgi and then subsequently secreted to the extracellular space. The GFP-FM4-hGH construct has a furin cleavage site between FM domains and hGH and so the passage of cargo through the TGN, where furin is localized, can be monitored by furin cleavage of the product (Gordon et al, 2010). HeLa cells stably transfected with GFP-FM4-hGH were used and the transport was initiated by the addition of ligand. Cells were fixed at intervals of 5, 15, 25, and 40 min after release of cargo from the ER, and intra-Golgi transport was monitored by fluorescence microscopy. We found that the peak of the cargo moved from the *cis*- to *trans*-Golgi over time (Fig. 4A,B). This behavior was similar to VSVG-GFP-RUSH described earlier (Fig. 2A,B and Tie et al, 2016; Tie et al, 2022). Surprisingly when these cells were observed by EM, a clear presence of the protein could be seen in the TGN (Fig. 4C). Next, we monitored the secretion of GFP-FM4-hGH by biochemical methods (Fig. 4D). We found that the secretion of GFP-FM4-hGH to the extracellular medium started at around 20 min after exit from the ER and the peak was reached after 45–60 min after the ER exit. The secreted version of the protein was approximately 55 kDa (as monitored by anti-GFP antibody), while the intracellular version of the protein was around 75 kDa. The furin cleavage site at the junction of FM4 and hGH which when acted upon is expected to separate the hGH from GFP-FM4 leading to an approximate decrease of 22 kDa in molecular weight. This corresponds well with the results observed of the secreted version of the protein. Thus while fluorescence studies suggested that GFP-FM4-hGH did not apparently reach TGN before exit from the Golgi apparatus, EM and biochemical studies show that the protein had reached furin-containing TGN compartment and only those that had reached the TGN were secreted.

Thus, from the EM observations described in Figs. 3 and 4C where the cargoes were observed to reach the TGN, and the biochemical studies which show that only those cargoes that had reached the TGN were secreted (Fig. 4D), we conclude that transit through the TGN is an obligatory step during the transport of cargoes.

## TGN exit is a rate-limiting step in the transport of cargo

The above-described results demonstrate that the cargoes enter at the *cis*-Golgi, traverse the organelle to reach the TGN, and exit from there. If the exit of cargoes from TGN is coupled with their

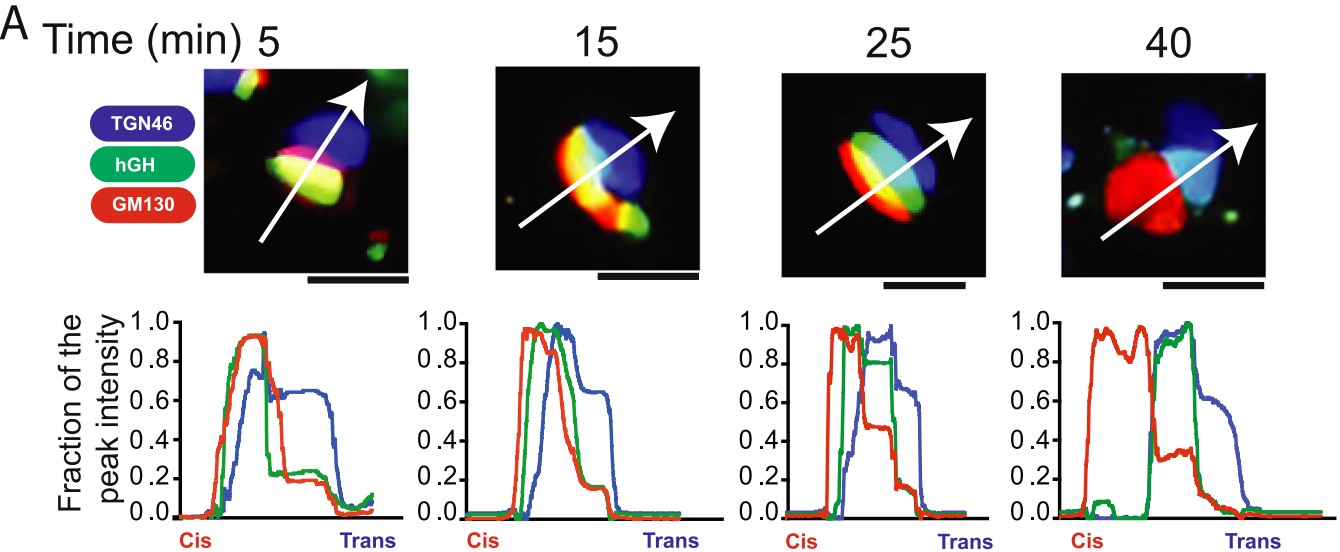

Normalized distance from origin

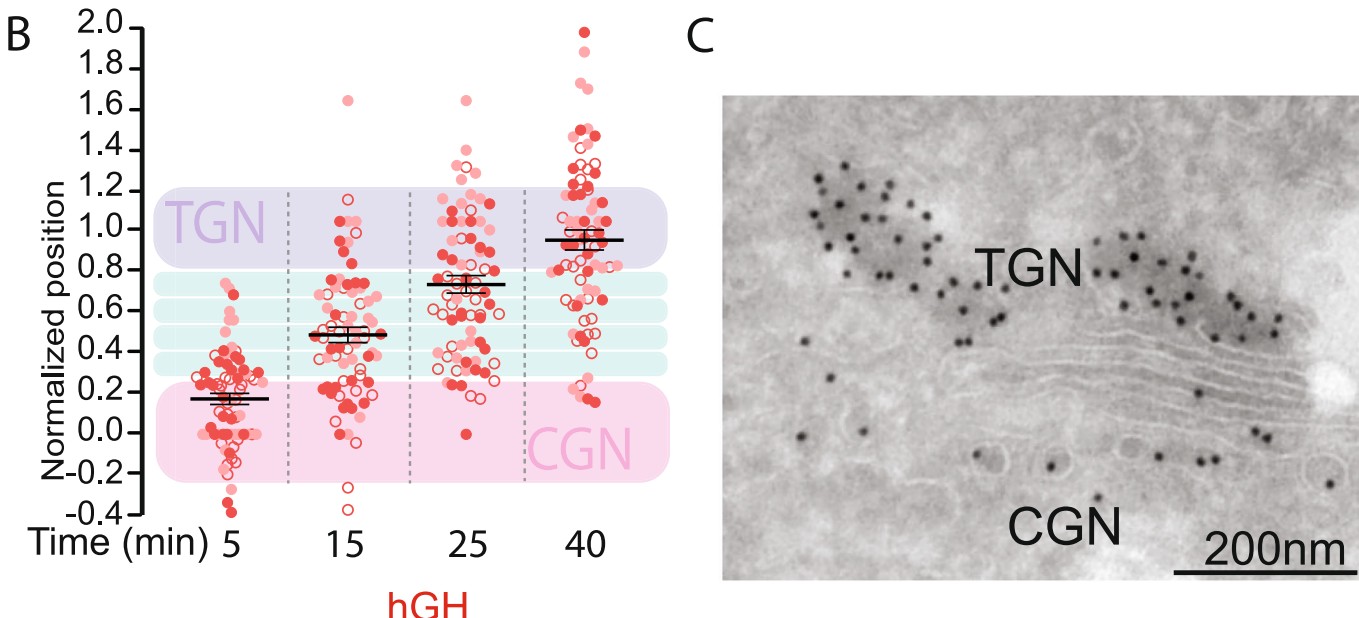

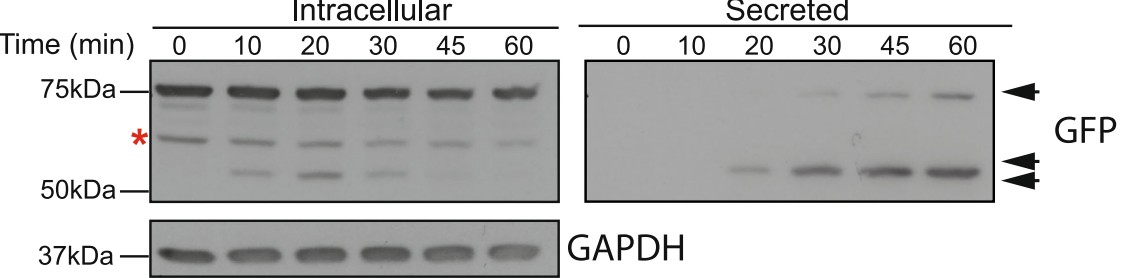

**Figure 4.  TGN exit is a rate-limiting step in cargo transport.**

(A) HeLa cells stably transfected with hGH-FM-GFP were treated with nocodazole (33 µM) for 3 h at 37 °C. Then they were incubated with DD solubilizer for the indicated times at 37 °C. The cells were fixed and stained for GM130 and TGN46 to mark the *cis*- and *trans*-Golgi, respectively. The position of GFP-FM4-hGH relative to GM130 and TGN46 is shown. Bar: 1 µm. (B) The position of the GFP-FM4-hGH and Golgi marker peaks was normalized, considering the position of the GM130 peak to be 0 and TGN46 to be 1. The relative position of GFP-FM4-hGH peaks over time is plotted (mean $+/-$ SE; Data points ($n > 70$) from three independent experiments are represented by grouped circles in three different shades of the same color). (C) Cells prepared as in (A) were fixed 20 min after the addition of DD solubilizer and prepared for EM. The sections were labeled with anti-GFP antibody (10 nm gold). (D) HeLa cells stably transfected with GFP-FM4-hGH were incubated with DD solubilizer for indicated times at 37 °C. The culture supernatant and cell lysate were collected and analyzed by Western blotting with anti-GFP antibody. While the intracellular version of hGH-FM-GFP was approximately at 75 kDa, the secreted form of the protein was consistently at ~55 kDa. Arrowhead indicates full-length form of GFP-FM4-hGH and double arrowhead indicates furin cleaved form. Asterisk indicates a likely non-specifically cleaved protein since it is not secreted. Source data are available online for this figure.

arrival from the *cis*-Golgi, the kinetics of exit from TGN is expected to be linear (Patterson et al, 2008). In contrast, the kinetics of exit from the Golgi was found to be monoexponential ((Patterson et al, 2008) and Fig. 1B)). Patterson and colleagues explained this by suggesting there is no intra-Golgi transport of cargoes per se but our results (Figs. 2 and 3) and other published observations (Tie et al, 2016; Tie et al, 2022) clearly show that cargoes do indeed undergo an intra-Golgi transport before their exit, even when using temperature-independent transport protocols. An alternative and potential explanation for a monoexponential exit of cargoes is that they are concentrated in the compartment from where they exit. Patterson et al did examine this aspect but they restricted their analysis only to the Golgi stacks, excluding the TGN from consideration, and did not find cargoes accumulating in the *trans*-Golgi (Patterson et al, 2008). On the contrary, our observations on the intra-Golgi transport of cargoes show that they accumulate at the TGN 25–40 min after exit from the ER (Figs. 3 and 4). So to study if cargoes accumulate in the TGN, we chose to study a time point of 60 min after the start of the ER exit of cargoes, when the arrival of newly synthesized cargoes from the ER to Golgi is likely to be nearly absent. At this time point, we examined the distribution of temperature-sensitive VSVG-GFP by EM. Our results show that VSVG-GFP colocalizes with the TGN marker TGN46 (Fig. 5A). The observation of the colocalization of temperature-sensitive VSVG with TGN46 has also been made by others earlier using fluorescence microscopy (Dickson et al, 2020). Of note, the concentration of VSVG-GFP in the TGN is much less than what we had earlier observed with hGH-GFP, suggesting that cargo-specific mechanisms of concentration are also at play here. We next studied whether this was also the case with temperature-independent synchronization methods. So we studied the localization of the following RUSH constructs: VSVG-GFP (basolateral cargo), GPI-GFP (apical cargo), and LAMP1-GFP (lysosomal cargo). The peak of all of these cargoes coincided with that of TGN46 60 min after their exit from the ER (Fig. 5B,C). Accumulation of the cargoes in the TGN of the ministacks suggested that the cargoes will exit the Golgi in a monoexponential manner even from the ministacks (earlier we had observed a monoexponential exit from the Golgi ribbon, Fig. 1D). We tested this prediction with an iFRAP experiment using the E-cadherin-GFP-RUSH and found that indeed the cargoes exited in a monoexponential manner (Fig. 5D,E), confirming their accumulation in the TGN compartment under this condition. To understand if this observation also holds true in vivo, we examined the distribution of very low-density lipoprotein (VLDL) particles in the Golgi apparatus of liver tissue sections (Fig. EV4A–F). We found that electron-dense VLDL particles were mostly present in the TGN and were concentrated

there. This is in line with earlier observations that most large cargoes (like procollagen-I) (Beznoussenko et al, 2022; Weinstock and Leblond, 1974) and small diffusible cargoes like albumin and anti-trypsin (Beznoussenko et al, 2022; Beznoussenko et al, 2014) are seen accumulated in the TGN at steady state. These results in sum suggest that exit from the TGN is a rate-limiting step in the transport of cargoes across the secretory pathway leading to their accumulation at the TGN and this accumulation can explain the exponential exit of cargoes from the Golgi area.

## Discussion

Taken together, the results described above show that: 1) newly arrived cargoes progress sequentially from *cis*- to *medial* and *trans*-Golgi to TGN and 2) accumulate in the TGN from where they eventually exit at monoexponential rates independent of transport rates through the stack. Thus, these results support models that involve vectorial transport across the Golgi stack (Beznoussenko et al, 2022; Bonfanti et al, 1998; Glick and Luini, 2011; Mironov and Beznoussenko, 2019). While these conclusions are inconsistent with an important prediction of the rapid partitioning model that there is no "intra-Golgi transport" of cargoes, they are not contrary to other propositions of the model. For instance, the observation that cargoes and Golgi residents segregate within the Golgi apparatus, (as also reported by others (Chen et al, 2017; Perinetti et al, 2009; White et al, 2001), likely based on lipids. Changes in the lipid composition of Golgi membranes are known to affect the kinetics of transport and sorting at the TGN (Kockx et al, 2012; Surma et al, 2012). Lipid-based sorting of Golgi proteins into export and processing domains is an interesting concept and based on our results here it may correspond to the Golgi stack and TGN (Jennifer Lippincott-Schwartz, personnel communication). While the role of lipid-based sorting for Golgi export is known (Surma et al, 2012; von Blume and Hausser, 2019), its relevance for the processing of cargoes in the Golgi remains unexplored. Integrating these aspects would be essential for any successful model of intra-Golgi transport.

The differences between the data reported here and those of Patterson et al in the presence of a flat or slow phase before the monoexponential exit of cargoes from the Golgi zone likely result from the differences in cell type and the experimental conditions including cargo load that were utilized. For instance, previous studies have shown that cargo waves reaching the *cis*-Golgi can induce the interconnections between Golgi cisternae that would promote faster intra-Golgi transport of cargoes by diffusion and thus may diminish the flat/slow phase (Marsh et al, 2004; Mavillard et al, 2010; Trucco et al, 2004).

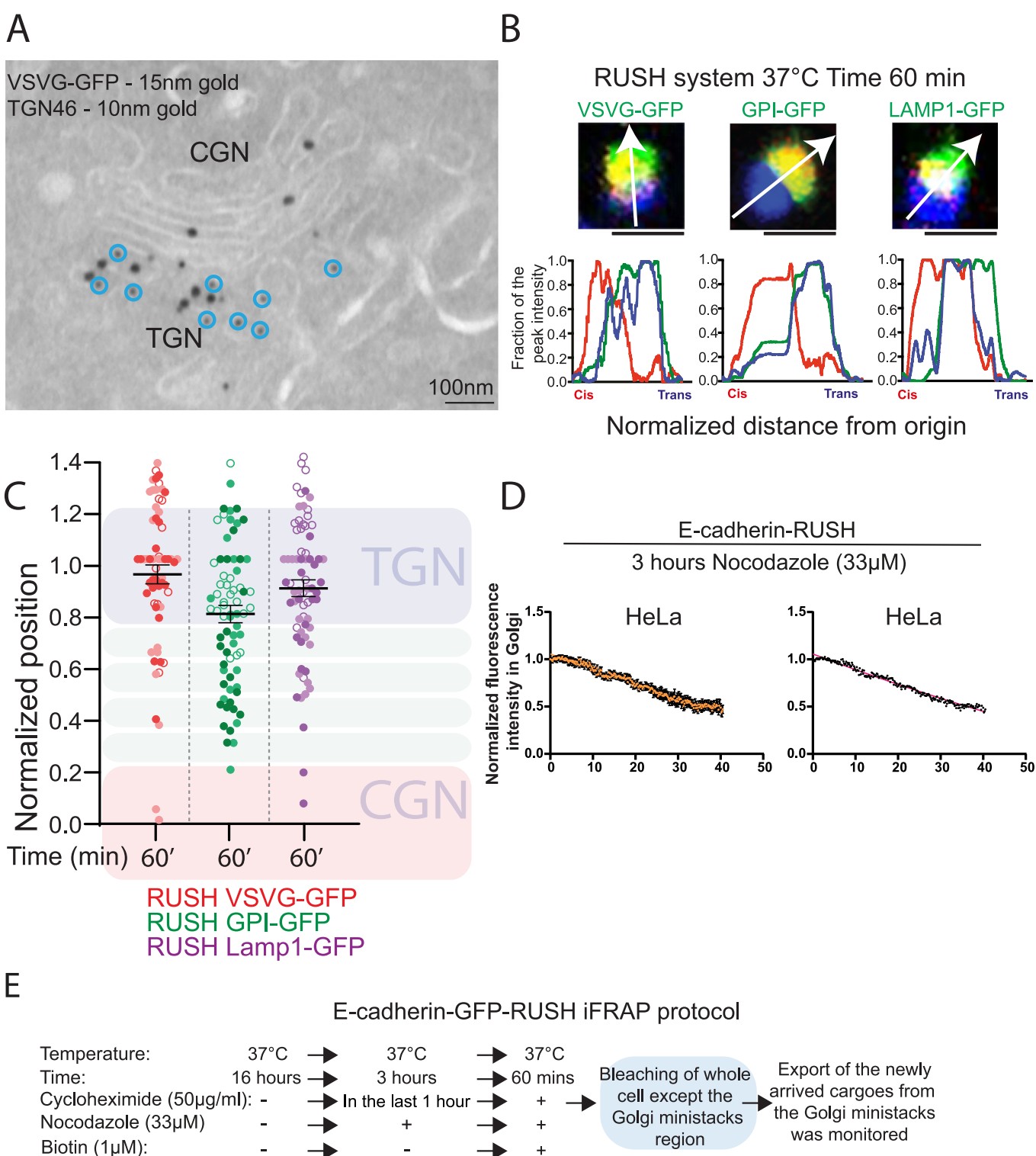

**A**

VSVG-GFP - 15nm gold
TGN46 - 10nm gold

CGN

TGN

100nm

**B**

RUSH system 37°C Time 60 min

VSVG-GFP    GPI-GFP    LAMP1-GFP

Fraction of the peak intensity

Cis    Trans    Cis    Trans    Cis    Trans

Normalized distance from origin

**C**

Normalized position

TGN

CGN

Time (min)    60'    60'    60'

RUSH VSVG-GFP
RUSH GPI-GFP
RUSH Lamp1-GFP

**D**

E-cadherin-RUSH

3 hours Nocodazole (33µM)

HeLa                    HeLa

Normalized fluorescence intensity in Golgi

**E**

E-cadherin-GFP-RUSH iFRAP protocol

| Temperature: | 37°C → | 37°C → | 37°C |
|---|---|---|---|
| Time: | 16 hours → | 3 hours → | 60 mins |
| Cycloheximide (50µg/ml): | - → | In the last 1 hour → | + |
| Nocodazole (33µM): | - → | + → | + |
| Biotin (1µM): | - → | - → | + |

Bleaching of whole cell except the Golgi ministacks region → Export of the newly arrived cargoes from the Golgi ministacks was monitored

Tie et al suggested that cargoes moving vectorially across the Golgi apparatus exit from the *trans*-Golgi part of the Golgi stack and not from the TGN (Tie et al, 2016; Tie et al, 2022). However, there are some limitations to these studies and the protocols used. First, is the resolution limit of IF-based methods in general, especially when dealing with the convoluted and complicated Golgi stack. Indeed, because of this limitation, we decided to examine intra-Golgi transport by complementary methods of EM and biochemical methods. EM analysis provides very high resolution to clearly distinguish between the Golgi stack and the associated vesicular tubular clusters. Further, by identifying the TGN based on morphology rather than markers it overcomes inherent limitations

**Figure 5.  TGN is a rate-limiting step in the transport of cargoes through the secretory pathway.**

(A) HeLa Cells were transfected with VSVG-GFP and kept overnight at 40 °C. Then cells were shifted to 32 °C for 1 h and then fixed and prepared for EM. The sections were labeled for GFP (15 nm gold) and TGN46 (10 nm gold; indicated by blue circles). Bar: 100 nm. (B) HeLa Cells were transfected with VSVG-GFP-RUSH, GPI-GFP-RUSH or LAMP1-GFP-RUSH constructs and kept at 37 °C. Then cells were incubated with 1 µM biotin for 60 min at 37 °C. The cells were fixed and stained for GM130 and TGN46 to mark the *cis*- and *trans*-Golgi, respectively. The relative position of the cargo relative to the GM130 and TGN46 is shown. Bar: 2 µm. (C). In HeLa cells treated as in (B), the position of each indicated RUSH construct and Golgi marker peaks were normalized, considering the position of the GM130 peak to be 0 and TGN46 to be 1. The relative position of each indicated RUSH construct peak at 60 min is plotted (mean $+/-$ SE; Data points ($n > 70$) from three independent experiments are represented by grouped circles in three different shades of the same color). (D) HeLa cells were transfected with E-cadherin-GFP-RUSH constructs overnight and then treated with nocodazole (33 µM) for 3 h at 37 °C. The cells were treated with cycloheximide for the last 1 h at 37 °C. Biotin was then added to cells to allow the cargoes to enter the Golgi ministacks for 60 min. Then the fluorescence in the cell, except for a few Golgi ministacks, was photobleached (iFRAP). Left panel: the exit of the cargoes from the Golgi ministack was then monitored, and the fluorescence in the Golgi area was calculated over time and represented as normalized to the initial (after iFRAP) fluorescence levels (mean $+/-$ SE; $n = 2$; 5 ministacks). Right panel: the data for the E-cadherin-GFP-RUSH exiting from Golgi ministacks fit well with a monoexponential decay (red) with the following rate constants: 0.005839 min$^{-1}$, with an R-squared value > 0.97. (E) Schematic representation of (C) of the iFRAP protocol for E-cadherin in cells treated with nocodazole. Source data are available online for this figure.

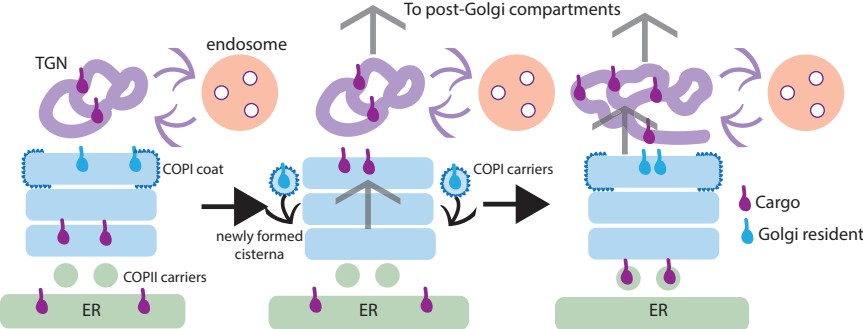

**Figure 6.   Model for *trans*-Golgi to TGN transport.**

Our proposed model of how cargoes are transported from the *trans*-Golgi to TGN. We propose that when cargoes reach the *trans*-Golgi cisterna, the residents and cargoes are separated by the action of COPI. The cargo-containing cisterna then interacts and fuses with the TGN network to deliver the cargo to the compartment. This in effect can be considered a "carrier" mediated transport of cargoes from *trans*-Golgi to TGN. The rate of input of cargoes into the compartment is more than the rate of their exit from the compartment, leading to their accumulation in the TGN. The TGN is a dynamically stable structure that depends on inputs from the Golgi stack as well as from the endosomes. An increase in either of these inputs leads to hypertrophy of the TGN as happens during increased cargo transport or during increased lysosomal biogenesis (Griffiths and Simons, 1986).

due to TGN subcompartmentalization. Our conclusion is further strengthened by the use of an orthogonal biochemical method (furin cleavage of cargo proteins) which demonstrates that cargoes indeed pass through morphologically and biochemically distinct TGN. Of note, furin almost entirely localizes to the TGN and the post-Golgi structures as observed by EM and functional assays (Bosshart et al, 1994; Molloy et al, 1994) and as also confirmed by Tie and colleagues (Tie et al, 2016; Tie et al, 2022). Further, Patterson and colleagues demonstrated that at least some of the VSVG molecules present in the Golgi apparatus are insensitive to Brefeldin A and remain in the perinuclear area while the Golgi enzyme (galactosyltransferase) returns to the ER (Patterson et al, 2008), suggesting that the cargo molecules were likely present in the TGN. Thus, we conclude that cargoes traverse the TGN as part of their intra-Golgi transport and exit the Golgi apparatus at the TGN. Of note, a TGN localization of cargoes is observed with basolateral (VSVG), apical (GPI-GFP), lysosomal (LAMP1-GFP) as well as cargoes that form "secretory granules" (GFP-FM4-hGH), underscoring that TGN is the compartment in which cargoes accumulate before being sorted to their correct destination within the cell.

We propose that the TGN is a distinct and rate-limiting compartment before the exit of cargoes. This is in line with the earlier proposals of TGN as a distinct compartment (Nakano, 2022; Tojima et al, 2019) and also

explains the observed differential transport kinetics of cargoes out of the Golgi apparatus (Boncompain et al, 2012; Boncompain and Perez, 2013). But this does not explain other observations of differential intra-Golgi transport rate (Tie et al, 2024). These could be because of faster intra-Golgi transport mediated by tubules (Beznoussenko et al, 2014) and or increased recycling of cargoes (Casler et al, 2019; Fossati et al, 2014). These suggest there is still a lot more to understand in pathways of intra-Golgi transport.

How are cargoes transported from *trans*-Golgi to TGN? Live imaging studies in yeast have shown that the *trans*-Golgi cisterna matures into TGN (Casler et al, 2019; Kurokawa et al, 2019). While this process cannot be directly observed in mammalian cells, the classical EM studies showing images of *trans*-cisternae that are "peeling off" (Clermont et al, 1995; Ladinsky et al, 1994) suggest a similar process also in mammalian cells. On the other hand, there is evidence that *trans*-Golgi to TGN transport in mammalian cells may be more complicated than simple maturation: (1) Block of the transport out of TGN leads to an accumulation of cargoes and enlargement of TGN with no increase in the cisternal number as would be expected from a maturation-based transport. (2) Maturation requires a recycling from distal to proximal compartments but in mammalian cells recycling from TGN to earlier compartments of the Golgi is limited (see (Mellman and Simons, 1992; van Deurs et al, 1988). (3) In vitro reconstitution study has

shown transport from *trans*-Golgi to TGN is likely mediated by cargo-containing carriers rather than maturation-based mechanisms (Pullikuth and Weidman, 2002).

Based on these considerations, we propose the following working model for the transport of cargo from *trans*-Golgi to TGN (Fig. 6): (1) as the cargo reaches the *trans*-Golgi, the *trans*-Golgi residents are removed by retrograde carriers leaving a *trans*-most cisterna (or carriers) containing only cargoes; (2) this emerging cargo-containing cisternal membrane fuses with the already existing TGN (Clermont et al, 1995) and thus is incorporated into the TGN compartment. (3) the cargo from the TGN is sorted into varied transport routes depending on the signals present on the cargo (Bonifacino, 2014; Gonzalez and Rodriguez-Boulan, 2009; Guo et al, 2014; Ramazanov et al, 2021; Stalder and Gershlick, 2020; von Blume et al, 2012; Wakana et al, 2012; Weisz and Rodriguez-Boulan, 2009). The cargo then exits the TGN compartment depending on the rate of formation and microtubule-dependent translocation of the post-Golgi carriers. The rate of formation of post-Golgi carriers is slower than the rate of arrival of cargoes to the TGN, which then gets incorporated with the pre-existing cargo-containing TGN, leading to cargo accumulation in the TGN. Their monoexponential rate of exit from TGN thus reflects only a single component i.e., the coupled formation and transport out of the Golgi area of post-Golgi carriers. This discontinuous mode of transport between *trans*-Golgi and TGN also provides a mechanistic explanation for the proposal that TGN can act as a directional valve to limit the flow of lipids and glycoproteins (Mellman and Simons, 1992) and thus allow the assembly of the cholesterol and sphingolipid-enriched lipid territories (Holthuis and Menon, 2014). Further, the potential to regulate the rate of exit of cargoes from the TGN (Malhotra and Campelo, 2011; preprint: Di Martino et al, 2020) provides a way to tune the rate-limiting nature of TGN exit to provide enough time for the completion of cargo glycosylation to appropriate levels (Nabi and Dennis, 1998).

To conclude we propose an intriguing model of transport through the Golgi apparatus wherein the cargo is transported through the stack in a maturation-based mechanism followed by a non-maturation-based mechanism (likely to be dissociative/vesicular transport or fusion-based transport) from the stack to TGN and beyond. This model is compatible with the polarized organization of the Golgi apparatus promoting sequential exposure of cargoes to glycosylation enzymes and also with distinct transport kinetics of cargoes with dissimilar Golgi residence time requirements for their efficient processing. Investigation of the molecular details of the processes involved in the transport from the Golgi stack to TGN would provide interesting insights into the functioning and evolution of the mammalian Golgi apparatus.

# Methods

### Reagents and tools table

| Reagent/resource | Reference or source | Identifier or catalog number |
| --- | --- | --- |
| **Experimental models** | | |
| HeLa-M cells | P. Pothukuchi et al, 2021 | N/A |
| HeLa-M hGH-FM-GFP cells | De Gordon et al, 2010 | N/A |
| A375MM cells | A. Filograna et al, 2024, kind gift from Dr. D. Corda | N/A |
| Cos7 cells | G. Grimaldi et al, 2022, kind gift from Dr. D. Corda | N/A |
| RPE cells | F. Mascanzoni et al, 2024, kind gift from Dr. A. Colanzi | N/A |
| 6-month-old male Wistar rats | Cardiological Centre in Moscow, Russia | N/A |
| **Recombinant DNA** | | |
| ts045-VSVG-GFP | Addgene | Cat #11912 |
| VSVG-GFP-RUSH | Addgene | Cat #65300 |
| GPI-GFP-RUSH | Addgene | Cat #65296 |
| VSVG-mCherry-RUSH | Addgene | Cat #65301 |
| LAMP1-GFP-RUSH | Kind gift from Dr. Franck Perez | N/A |
| E-cadherin-GFP-RUSH | Kind gift from Dr. Franck Perez | N/A |
| **Antibodies** | | |
| Monoclonal Mouse GM130 | BD Biosciences | Clone 35 Cat # 610822 RRID: AB_398141 |
| Polyclonal Sheep anti-human TGN46 | BioRad/AbD-Serotec | Cat #AHP500G RRID:AB_323104 |
| GOLGB1 (Giantin) | Invitrogen | PA5-52772 |
| Anti-mouse, donkey Alexa Fluor 568 | ThermoFisher Scientific | Cat #A10037 RRID:AB_2534013 |
| Anti-sheep, donkey Alexa Fluor 633 | ThermoFisher Scientific | Cat #A-21100 RRID:AB_2535754 |
| Monoclonal Mouse GFP | Abcam | Cat #ab6556 RRID:AB_305564 |
| Polyclonal Rabbit VSVG | Bethyl Laboratories Inc. | Cat #A190-131A; RRID:AB_155862 |
| Protein A gold 10 nm | Cell Microscopy Core, UMC Utrecht | N/A |
| Protein A gold 15 nm | Cell Microscopy Core, UMC Utrecht | N/A |
| **Oligonucleotides and other sequence-based reagents** | | |
| **Chemicals, enzymes, and other reagents** | | |
| TransIT-LT1 | Mirus Bio | Cat # MIR2304 |
| Lipofectamine LTX | Life Technologies | Cat # 15338100 |
| Paraformaldehyde 8% | Electron Microscopy Sciences | Cat # 157-8 |
| Glutaraldehyde 8% | Electron Microscopy Sciences | Cat # 16000 |
| Biotin | Sigma Aldrich | Cat # B4501 |
| Cycloheximide | Sigma Aldrich | Cat # 01810 |
| D/D solubilizer | Takara Bio | Cat # 635054 |
| Trichloroacetic acid | Sigma Aldrich | Cat # T6399 |
| Mini-Protean Pecast Gels | Biorad | Cat # 4568081 |
| ECL | GE Healthcare | Cat # RPN2209 |

| Reagent/resource | Reference or source | Identifier or catalog number |
|---|---|---|
| Complete mini EDTA free protease inhibitor cocktail | Roche | Cat # 4693159001 |
| **Software** | | |
| Zen software system | Carl Zeiss | |
| Metamorph 7.7.3.0 | Universal Imaging | |
| ImageJ | Open source | |
| Prism 8.0 | GraphPad | |
| **Other** | | |

## Reagents

All reagents and chemicals are listed in the Reagents and Tools Table. were of molecular biology grade. Lipofectamine 2000 (Cat #11668027) and Lipofectamine LTX with PLUS (Cat #15338100) were purchased from ThermoFisher Scientific, USA. DD solubilizer was purchased from Takarabio. TransIT-LT1 (Cat #MIR 2305) was purchased from Mirus, USA. RPMI-1640 (Cat #21875), DMEM (Cat #41965), DMEM/F-12 (Cat #11320033), and FBS (Cat #10437036) were purchased from Gibco/ThermoFisher Scientific, USA. Protein A gold 5 nm and Protein A gold 10 nm were acquired from Cell Microscopy Core, UMC Utrecht. Biotin (Cat #29129) was purchased from Pierce, USA. Bacterial strains E. coli (DH5α) (Cat #18265017) and E. coli (BL-21-DE3) (Cat #C600003) were purchased from ThermoFisher Scientific, USA.

## Cell lines

Cell lines utilized in this paper are listed in the Reagents and Tools Table. HeLa-M cell lines (see below) were cultured in RPMI-1640 supplemented with 10% FCS. Human melanoma cells A375MM and Cos7 cells were cultured in DMEM supplemented with 10% FBS. Retinal pigment epithelium (RPE) cells were grown in DMEM/F-12 supplemented with 10% FBS. All media were supplemented with 100 U/ml penicillin/streptomycin and 2 mM L-glutamine. All cells were grown in a controlled atmosphere (5% $CO_2$ and 95% air) at 37 °C. Mycoplasma contamination was not observed in cell cultures as detected using PCR assay. Cell cultures between 3 and 15 passages were used for the experiments, and the cells were cultured to 80% confluence for the experiments unless indicated otherwise.

## Plasmids and siRNA transfection

HeLa-M cells, RPE, A375, and COS7 cells were transfected with plasmid vectors using TransIT-LT1 or Lipofectamine LTX reagents according to manufacturer's instructions. A list of plasmids used in this study can be found in Reagents and Tools table.

## Immunofluorescence and confocal microscopy

For immunofluorescence analysis, cells were grown on coverslips and fixed in 4% paraformaldehyde (Electron Microscopy Sciences, Hatfield, USA) for 10 min at room temperature (RT). The cells were then permeabilized and incubated with blocking buffer (0.05% saponin and 0.5% BSA in PBS) for 10 min at RT followed by incubation with specific primary antibodies (see Reagents Tools Table) for 1 h at RT and washed with PBS. Cells were subsequently labeled with appropriate Alexa Fluor-conjugated secondary antibodies (see Reagents Tools Table). The coverslips were mounted using Mowiol, and images were acquired using the confocal microscope Zeiss LSM700.

## Line scan analysis

Confocal images were acquired under non-saturation conditions using a ×63 objective (1.4 NA). The line scan analysis was performed as described previously (Rizzo et al, 2013) using the Zen software system (Carl Zeiss). Only Golgi stacks with clearly separated GM130- and TGN46-stained zones were used for the analysis. A line was drawn in the middle of the stacks along the *cis–trans* direction, and the fluorescence intensity of each stained marker along this line was plotted. The distances were normalized by considering the start of the GM130 peak as 0, and the end of the TGN peak as 1. At least 25 stacks were examined per treatment. For each condition, a representative plot of data and an image of a Golgi stack processed using the "Image with Zoom" function of Metamorph 7.7.3.0 (Universal Imaging), are shown.

## GLIM method of quantitating intra-Golgi transport

The method was essentially as described in (Tie et al, 2017). After identifying the coordinates of the center of masses of GM130, TGN46 and VSVG-GFP, we calculated the distance between the GM130 and TGN46 as well as GM130 and VSVG-GFP using the formula:

$$d = \sqrt{((x\_GM130\text{-}x\_TGN46)^2 + (y\_GM130\text{-}y\_TGN46)^2)} \text{ or}$$

$$d = \sqrt{((x\_GM130\text{-}x\_VSVG)^2 + (y\_GM130\text{-}y\_VSVG)^2)},$$

respectively.

Then the distance between the VSVG-GFP and the line connecting GM130 and TGN46 was calculated using the formula:

$$D = l((y\_GM130\text{-}y\_TGN46)x\_VSVG)$$
$$- ((x\_GM130\text{-}x\_TGN46)y\_VSVG)$$
$$+ x\_TGN46^*y\_GM130\text{-}x\_GM130^*y\_TGN46l/$$
$$\sqrt{((x\_GM130\text{-}x\_TGN46)^2 + (y\_GM130\text{-}y\_TGN46)^2)}$$

The distance from GM130 to the intersection of this line with the line connecting GM130 and TGN46 (essentially the distance moved by VSVG-GFP within the Golgi compared to GM130) was calculated using the Pythagorean function.

## Analysis of carriers

Analysis of the number of carriers was done using ImageJ. The area containing the central Golgi was excluded from the analysis. The carriers were selected by intensity thresholding and they were counted in each frame using the Analyze particle function set to count particles ranging from 50 to infinity-sized pixels. The total number of particles were normalized to the highest value and plotted.

## Immunogold labeling of cells

For immunogold labeling, samples were processed essentially as described earlier (Rizzo et al, 2013). In brief, cells were fixed with 2% formaldehyde and 0.2% glutaraldehyde in PHEM buffer (0.1 M), processed to be embedded in gelatin and 50-nm sections were cut with a diamond knife on a UC7 Leica cryo-ultramicrotome. The sections were incubated with the rabbit anti-GFP antibody followed by Protein A gold (10 nm) to reveal antigen staining. When the anti-VSVG monoclonal antibody or anti-TGN46 antibody was used, then sections were incubated with anti-mouse or anti-sheep antibodies raised in Rabbit before incubation with Protein A gold. After labeling, the sections were treated with 1% glutaraldehyde and embedded in methylcellulose uranyl acetate, followed by analysis in a 120 kV FEI Tecnai 12 Biotwin electron microscope (FEI, Eindhoven, The Netherlands) using a VELETA digital camera. The polarity of the Golgi stacks was defined by compositional (*cis*-Golgi marker GM130 or TGN marker TGN46) or morphological parameter (perforated cisterna indicated *cis*-Golgi and clathrin buds indicated *trans*-Golgi). For quantitation, the perforated *cis*-most cisterna labeled with GM130 was considered C1 and the last cisterna, usually the C4, represented the *trans*-Golgi. Most of the Golgi stacks had four cisternae and rarely they had either three or more than four cisternae. TGN was defined as the area in front of the *trans*-Golgi cisterna (C4) consisting of tubular-vesicular elements and rarely cisternal fragments up to a distance that equals the thickness of the Golgi stack. The distribution of cargoes in the Golgi stack was expressed as the fraction of gold particles in each cisterna of the Golgi stack and TGN.

## EM analysis of tissue samples

Samples for analysis of the post-Golgi compartments in liver were taken from rats used for our previous studies (Sesorova et al, 2020; Sesorova et al, 2022). Briefly, animals were anaesthetized by ether and while the animals were under ether anesthesia, the abdomen of the animal was opened, and the tissue samples were removed for processing. Before the end of anesthesia the animals were sacrificed by a trained professional by the intracranial administration of a saturated solution of potassium chloride at a dose of 1–2 mM/kg (Sesorova et al, 2020). Death was confirmed by observing cessation of heartbeat and respiration, and absence of reflexes, in agreement with international standards (https://www.lal.org.uk. access date: 20 March 2022). The samples that were removed, processed, embedded, sectioned and stained as described previously (Sesorova et al, 2020; Sesorova et al, 2022).

All experimental animal procedures were approved by the Committees of the Ivanovo State Medical Academy and St. Petersburg State Pediatric University. The procedures for animal use were conducted in accordance with the ethical and legal standards of the Russian Federation mentioned in Order no. 755 of the Ministry of Health of the USSR on 12 August 1977. "On measures to further improve the organizational forms of work using experimental animals" and a letter from the Ministry of Agriculture dated 5 February 2022 no. 13-03-2/358, "On modern alternatives to the use of animals in the educational process" and 2010/63/EU legislation on animal protection. The experiments were approved by the decision of the Academic Council of St. Petersburg Pediatric University no. 10 from 23

September 2015 and the decision of ethic committee of Ivanovo State Medical Academy (no. 1 from 5/XII, 2018) in compliance with the above-mentioned Order no. 755 of the Ministry of Health of the USSR on 12 August 1977. "On measures to further improve the organizational forms of work using experimental animals" and a letter from the Ministry of Agriculture dated 5 February 2022 no. 13-03-2/358, "On modern alternatives to the use of animals in the educational process". All experiments on live animals were carried out in Russia; samples irreversibly fixed with glutaraldehyde, embedded in Epon or gelatin (with subsequent fixation) in Russia and only then were plastic samples transported to Italy, where these plastic samples were examined.

## iFRAP assay

Cells were plated on 35-mm glass-bottom microwell dishes (MatTeK, USA) and transfected with VSVG-GFP for 16 h at 40 °C or E-Cadherin-GFP-RUSH for 16 h at 37 °C. The cells were shifted to a microscope (Zeiss LSM700 or LSM 980 confocal microscope), where the stage temperature was set to 32 °C or 37 °C and 5% $CO_2$ for 30 min. Then the Golgi region was photobleached (100 bleaching iterations; 100% laser power) and the fluorescence was allowed to recover for 5 min. Immediately, the whole cell except the Golgi region was photobleached and the fluorescence intensity in the Golgi region was monitored by imaging every 30 s for 40–60 min (2% laser power). Fluorescence in the Golgi region was measured and presented, normalized to the Golgi intensity immediately post-bleaching.

## Experimental conditions for RUSH experiments

Cells were transiently transfected with RUSH constructs at 37 °C in the absence of biotin for 16 h, to localize the proteins to the endoplasmic reticulum (ER). Their ER exit was promoted by the addition of biotin (40 μM) for indicated times.

## Secretion of hGH-FM-GFP

Briefly, HeLa GFP-FM4-hGH were plated in a six-well plate (250,000 cells/well). The following day, complete media was replaced with media without serum plus cycloheximide (50 μg/ml). 30 min after the addition of cycloheximide, hGH secretion was initiated by adding DD solubilizer for the indicated time points. Conditioned media and cell lysates were collected for each time point and then conditioned media was TCA precipitated and analyzed by SDS-PAGE.

## Statistics

Error bars correspond to either standard deviation (SD) or standard error of mean (SEM) as indicated in figure legends. Statistical evaluations were done using unpaired two-tailed Student's *t* tests), and significance values are all marked as follows *$P < 0.05$, **$P < 0.01$, and ***$P < 0.001$ (ns, not significant).

# Data availability

This study includes no data deposited in external repositories.

The source data of this paper are collected in the following database record: biostudies:S-SCDT-10_1038-S44319-025-00548-9.

## Peer review information

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

## Acknowledgements

The authors thank Roman Polishchuk for the suggestion to consider TGN as distinct from the Golgi stack. Dr. I. Sesorova is acknowledged for her help in work with animals and the preparation of these samples. We thank the Bioimaging Facility of IEOS-CNR for access to technologies. We thank MEDINTECH, the Italian Node of Euro-Bioimaging (Preparatory Phase II –INFRADEV), Italian Cystic Fibrosis Research Foundation (FFC #6-2019), POR Campania project 2014-2020 (C.I.R.O.), SEELIFE and CTC project for financial support. AL acknowledges financial support from the AIRC (Projects IG 15767 and IG 20786), TERABIO, PON-IMPARA, and S.A.T.I.N. SP acknowledges financial support from the MUR (PRIN2022PNRR-2022MZJR9X, P2022YA3LL). DR acknowledges financial support from the MUR (PRIN2022PNRR P2022MMPXH) and MUR PNRR "National Center for Gene Therapy and Drugs based on RNA Technology" (Project no. CN00000041 CN3 RNA). IA and RDM were supported by the National Recovery and Resilience Plan (PNRR) project "SEE LIFE- StrEngthEning the ItaLIan InFrastructure of Euro-bioimaging".

## Author contributions

**Marinella Pirozzi**: Conceptualization; Investigation; Writing—original draft; Writing—review and editing. **Ilenia Agliarulo**: Conceptualization; Investigation. **Rosaria Di Martino**: Investigation. **Vincenzo Manuel Marzullo**: Investigation. **Micaela Quarto**: Investigation. **Gabriele Turacchio**: Investigation. **Galina V Beznoussenko**: Investigation. **Domenico Russo**: Conceptualization; Investigation; Writing—original draft; Writing—review and editing. **Alberto Luini**: Conceptualization; Formal analysis; Supervision; Funding acquisition; Writing—original draft; Project administration; Writing—review and editing. **Alexander A Mironov**: Conceptualization; Investigation; Writing—original draft; Writing—review and editing. **Seetharaman Parashuraman**: Conceptualization; Investigation; Writing—original draft; Project administration; Writing—review and editing.

Source data underlying figure panels in this paper may have individual authorship assigned. Where available, figure panel/source data authorship is listed in the following database record: biostudies:S-SCDT-10_1038-S44319-025-00548-9.

## Disclosure and competing interests statement

The authors declare no competing interests.

# Expanded View Figures

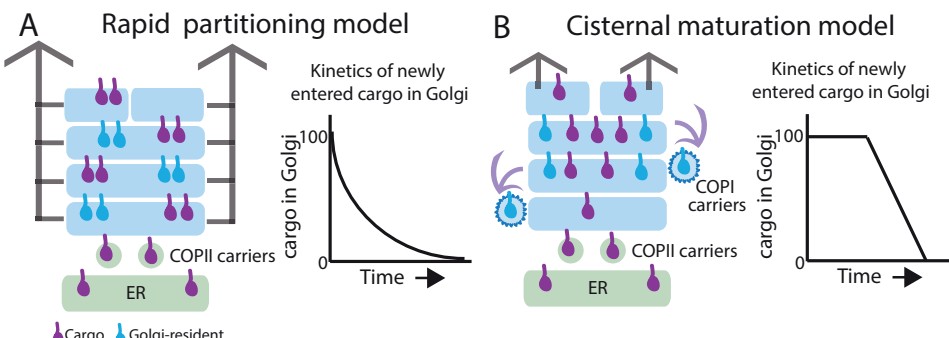

**Figure EV1. Comparison of Rapid partitioning and Cisternal maturation models.**

The comparison of rapid partitioning (**A**) and cisternal maturation model (**B**) of cargo transport through the Golgi. In the rapid partitioning model, the cargo that enters the Golgi distributes among the cisternae and leaves the Golgi from all the cisternae with no preferential site of exit. The amount of cargo exiting the Golgi depends on the amount of cargo present in the Golgi thus resulting in a monoexponential exit with no delay. Of note, in this model, there is no preferential exit of older or newly arrived cargoes from the Golgi. In the cisternal maturation model the cargoes remain in the cisterna that matures from *cis* to *trans*-Golgi. The cargoes thus arriving at the TGN (by the maturation of the *trans*-Golgi to TGN) is sorted into carriers that mature from the TGN to leave the Golgi. So, the kinetics of exit of newly arrived cargoes at the Golgi displays a flat or stationary phase that corresponds with the intra-Golgi transport and then linear kinetics of exit corresponding to the exit of the cargo. In this model, the older cargoes preferentially exit the Golgi first then followed by the newly arrived cargoes.

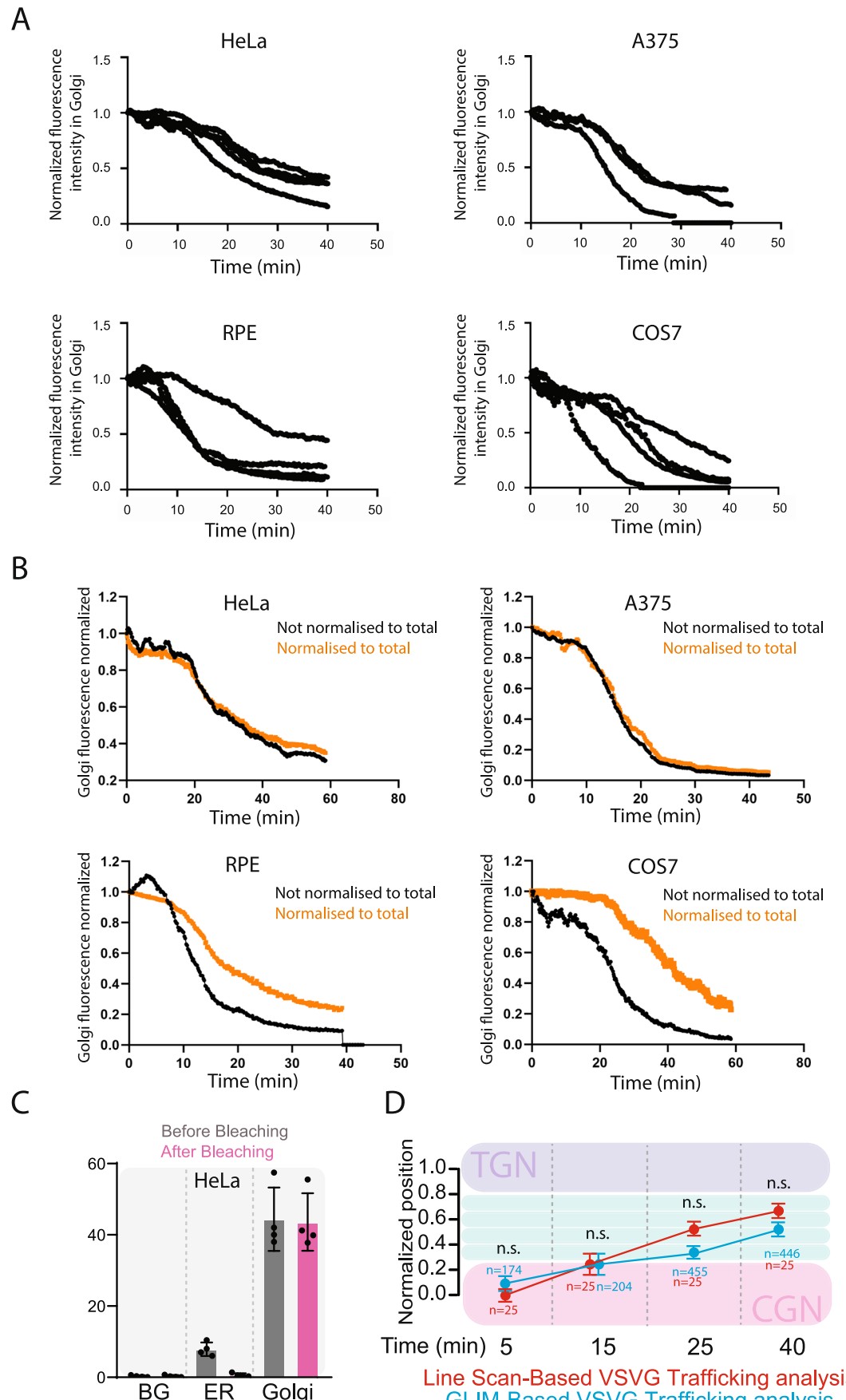

◀ **Figure EV2. iFRAP analyses of cargo exit from the Golgi apparatus.**

(A) The individual traces of the iFRAP data presented in Fig. 1B are shown. (B) Representative traces from A were normalized in two different ways. As presented in (A) or they were normalized by total fluorescence to account for bleaching (orange). The curves were qualitatively similar with flat phase followed by an exponential decrease. (C) The intensity levels of background (BG; area with no cells), ER and Golgi were measured before and after bleaching in the iFRAP protocol. The bleaching was efficient in the ER while there was little to no change in the Golgi fluorescence (mean ± SD; $n > 3$). (D) VSVG-GFP-RUSH construct localization with Golgi markers (GM130 and TGN46) from the replicate 2 of Fig. 2B. The positions of VSVG-GFP-RUSH and Golgi marker peaks were normalized, setting GM130 as 0 and TGN46 as 1. Measurements obtained using line scan analysis (red) are compared with those from the GLIM based automated method (blue) within the same experiment. Each data point represents an individual measurement, with results from both methods shown side-by-side for direct comparison. The relative position of the VSVG-GFP-RUSH construct is plotted at each indicated time point (median ± SE; $n$ is indicated in the figure, n.s = not significant; [Student's $t$ test]). Source data are available online for this figure.

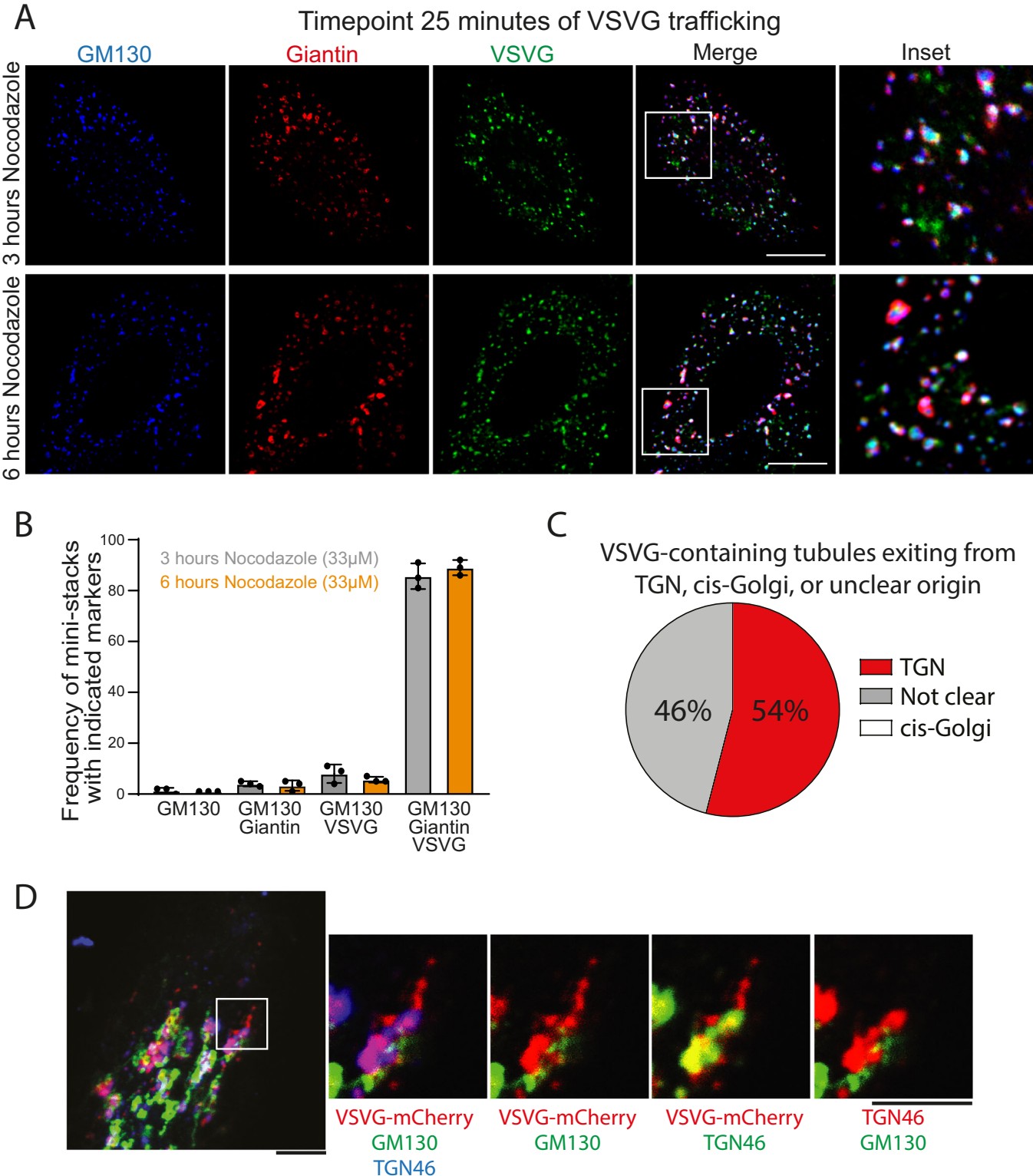

A — Timepoint 25 minutes of VSVG trafficking

GM130 Giantin VSVG Merge Inset

3 hours Nocodazole

6 hours Nocodazole

B

Frequency of mini-stacks with indicated markers

3 hours Nocodazole (33μM)
6 hours Nocodazole (33μM)

GM130 GM130 Giantin GM130 VSVG GM130 Giantin VSVG

C — VSVG-containing tubules exiting from TGN, cis-Golgi, or unclear origin

46% 54%

TGN
Not clear
cis-Golgi

D

VSVG-mCherry GM130 TGN46 VSVG-mCherry GM130 VSVG-mCherry TGN46 TGN46 GM130

◀

**Figure EV3. Validity of nocodazole treatment and exit of cargoes from the TGN.**

(A) HeLa cells were transfected with VSVG-GFP and kept overnight at 40 °C. Then cells were treated with nocodazole for the indicated amount of time and shifted to 32 °C for 25 min and then fixed and prepared for immunofluorescence with the indicated markers. White boxes represent insets. Bar: 10 μm. (B) Quantification from (A) of the number of ministacks showing the indicated markers. Mean +/− SD. Data points ($n > 290$) from 3 independent experiments. (C) The Pie chart shows the percentage of VSVG-mCherry-containing tubules exiting from TGN and those whose origin is unclear ($n = 13$). We have not observed any tubule originating from *cis*-Golgi. (D HeLa cells were transfected with VSVG-mCherry-RUSH construct overnight and then biotin was added for 1 h. The cells were fixed and stained for GM130 and TGN46 to mark the *cis*-Golgi and TGN respectively. White boxes represent insets. Bar: 5 μm. Source data are available online for this figure.

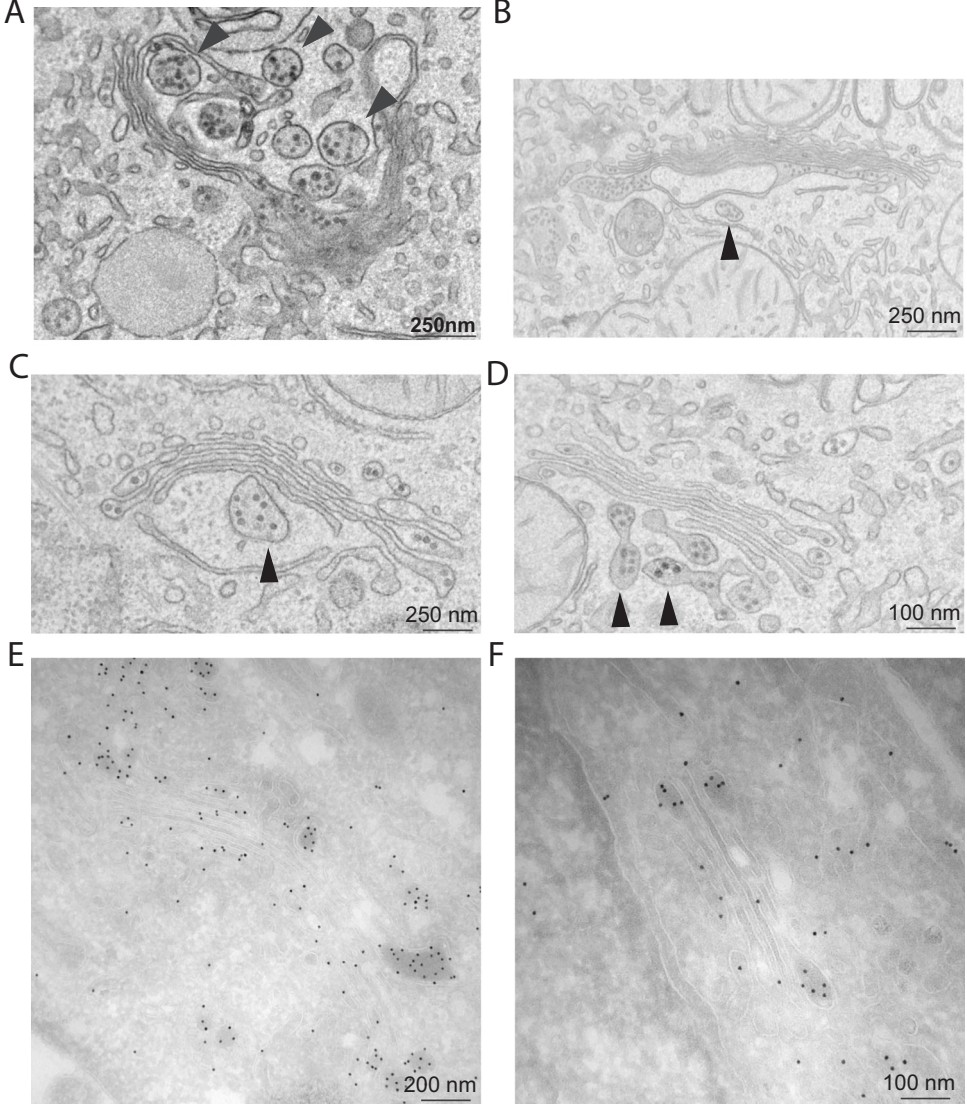

**Figure EV4. Cargoes accumulate in the TGN before exiting the Golgi apparatus.**

(**A**) Accumulation of VLDL particles in the TGN area of the hepatic tissue is indicated by black arrowheads. Bar: 250 nm. (**B–D**) Samples images showing the accumulation of VLDL particles in the TGN area of the hepatic tissue (indicated by black arrowheads as in Fig. EV4A). (**E, F**) Sample images from the same experiment of Fig. EV4A. Gold particles indicate GFP-FM4-hGH. Source data are available online for this figure.

