## [Peer Review File · EMBO Reports]

Cargoes move from cis to trans-Golgi compartments and concentrate in the TGN before exiting

Marinella Pirozzi, Ilenia Agliarulo, Rosaria Di Martino, Vincenzo Manuel Marzullo, Micaela Quarto, Gabriele Turacchio, Galina Beznoussenko, Domenico Russo, Alexandre Mironov, Seetharaman Parashuraman, and Alberto Luini

Corresponding author(s): Seetharaman Parashuraman (raman.sp@cnr.it), Alexandre Mironov (alexandre.mironov@ifom.eu), Marinella Pirozzi (m.pirozzi@ieos.cnr.it), Domenico Russo (d.russo@ieos.cnr.it), Alberto Luini (a.luini@ieos.cnr.it)

Review Timeline:

Submission Date:	19th Sep 24
Editorial Decision:	18th Nov 24
Revision Received:	15th Apr 25
Editorial Decision:	12th Jun 25
Revision Received:	20th Jun 25
Accepted:	10th Jul 25

Transaction Report:

Dear Dr. Mironov

Thank you for the submission of your research manuscript to our journal. We have now received the full set of referee reports that is copied below.

As you will see, the referees acknowledge that the findings are interesting and that your work contributes to reconciling different models of ER-Golgi trafficking based on new and enhanced technology. That said, the referees also have several suggestions how to further strengthen your work that I kindly ask you to address. A comparison to transport kinetics in live cells would strengthen the conclusions obtained with mini-stacks, if possible, but I am happy to discuss the suggestion on electron microscopy, which seems time intensive, also via a video chat, if you like.

Given these constructive comments, we would like to invite you to revise your manuscript with the understanding that the referee concerns (as detailed above and in their reports) must be fully addressed and their suggestions taken on board. Please address all referee concerns in a complete point-by-point response. Acceptance of the manuscript will depend on a positive outcome of a second round of review. It is EMBO Reports policy to allow a single round of revision only and acceptance or rejection of the manuscript will therefore depend on the completeness of your responses included in the next, final version of the manuscript.

We realize that it is difficult to revise to a specific deadline. In the interest of protecting the conceptual advance provided by the work, we recommend a revision within 3 months (February 18, 2025). Please discuss the revision progress ahead of this time with the editor if you require more time to complete the revisions.

I am also happy to discuss the revision further via e-mail or a video call, if you wish.

*****IMPORTANT NOTE:

We perform an initial quality control of all revised manuscripts before re-review. Your manuscript will FAIL this control and the handling will be delayed IN CASE the following APPLIES:

- 1) A data availability section providing access to data deposited in public databases is missing. If you have not deposited any data, please add a sentence to the data availability section that explains that.
- 2) Your manuscript contains statistics and error bars based on $n=2$. Please use scatter blots in these cases. No statistics should be calculated if $n=2$.

When submitting your revised manuscript, please carefully review the instructions that follow below. Failure to include requested items will delay the evaluation of your revision.*****

- 1) a .docx formatted version of the manuscript text (including legends for main figures, EV figures and tables). Please make sure that the changes are highlighted to be clearly visible.
- 2) individual production quality figure files as .eps, .tif, .jpg (one file per figure). Please download our Figure Preparation Guidelines (figure preparation pdf) from our Author Guidelines pages <https://www.embopress.org/page/journal/14693178/authorguide> for more info on how to prepare your figures.
- 3) a .docx formatted letter INCLUDING the reviewers' reports and your detailed point-by-point responses to their comments. As part of the EMBO Press transparent editorial process, the point-by-point response is part of the Review Process File (RPF), which will be published alongside your paper.
- 4) a complete author checklist, which you can download from our author guidelines (). Please insert information in the checklist that is also reflected in the manuscript. The completed author checklist will also be part of the RPF.
- 5) Please note that all corresponding authors are required to supply an ORCID ID for their name upon submission of a revised

manuscript (). Please find instructions on how to link your ORCID ID to your account in our manuscript tracking system in our Author guidelines
()

6) We replaced Supplementary Information with Expanded View (EV) Figures and Tables that are collapsible/expandable online. A maximum of 5 EV Figures can be typeset. EV Figures should be cited as 'Figure EV1, Figure EV2' etc... in the text and their respective legends should be included in the main text after the legends of regular figures.

7) Please note that a Data Availability section at the end of Materials and Methods is now mandatory. In case you have no data that requires deposition in a public database, please state so instead of refereeing to the database. See also < <https://www.embopress.org/page/journal/14693178/authorguide#dataavailability>>. Please note that the Data Availability Section is restricted to new primary data that are part of this study.

Additional information on source data and instruction on how to label the files are available .

10) Figure legends and data quantification:
The following points must be specified in each figure legend:

- the name of the statistical test used to generate error bars and P values,
 - the number (n) of independent experiments (please specify technical or biological replicates) underlying each data point,
 - the nature of the bars and error bars (s.d., s.e.m.)
- If the data are obtained from n {less than or equal to} 5, show the individual data points in addition to the SD or SEM.
- If the data are obtained from n {less than or equal to} 2, use scatter blots showing the individual data points.

See also the guidelines for figure legend preparation:
<https://www.embopress.org/page/journal/14693178/authorguide#figureformat>

11) Our journal encourages inclusion of *data citations in the reference list* to directly cite datasets that were re-used and obtained from public databases. Data citations in the article text are distinct from normal bibliographical citations and should directly link to the database records from which the data can be accessed. In the main text, data citations are formatted as follows: "Data ref: Smith et al, 2001" or "Data ref: NCBI Sequence Read Archive PRJNA342805, 2017". In the Reference list, data citations must be labeled with "[DATASET]". A data reference must provide the database name, accession number/identifiers and a resolvable link to the landing page from which the data can be accessed at the end of the reference. Further instructions are available at .

12) All Materials and Methods need to be described in the main text using our 'Structured Methods' format. According to this format, the Methods section includes a Reagents and Tools Table (listing key reagents, experimental models, software and relevant equipment and including their sources and relevant identifiers) followed by a Methods and Protocols section describing the methods, ideally using a step-by-step protocol format. The aim is to facilitate adoption of the methodologies across labs. Please download and fill our Reagents and Tools Table template (.docx), which you can find in our author guidelines: <https://www.embopress.org/page/journal/14693178/authorguide#structuredmethods>.

13) As part of the EMBO publication's Transparent Editorial Process, EMBO Reports publishes online a Review Process File to accompany accepted manuscripts. This File will be published in conjunction with your paper and will include the referee reports, your point-by-point response and all pertinent correspondence relating to the manuscript.

Yours sincerely,

=====

Referee #1:

Pirozzi and Agliarulo et al. have used a combination of classical cell biology and modern cell biology techniques and approaches to revisit the well-worked problem of intra-Golgi transport. In particular they take advantage of modern methodologies and thinking to re-examine an influential manuscript from the Lippincott-Schwartz laboratory in which a bi-phasic model of Golgi sorting is proposed. This is a complicated and controversial field which is marked by serious disagreement. It is therefore important that any new contributions contribute seriously to our understanding and are backed by strong evidence, otherwise they run the risk of further complicating it. With that in mind, I think that this manuscript does indeed make a worthy contribution due to a set of well thought out experiments and conclusions. I have several points that I think should be addressed, and I think it is important (for the above reasons) that they are addressed properly.

One of the issues in the manuscript to my mind is the switching from tsVSV-G to RUSH. It is well justified in the manuscript, but I think due to the absolute importance of comparable findings the experimental modality used in figure 1 should be used for the RUSH system.

In figures 2 and 4 the experimenters use a version of the GLIM methodology pioneered by the Lu lab. The authors are clearly aware of these contributions as they discuss them at length. There is a weakness in the analytical approach employed by this manuscript vs the work performed in the Lu lab, as the Lu approach models the Golgi ministack as a 2D gaussian rather than a line which results in higher quality centroid fitting. I encourage the authors to re-examine their data here and use this stronger analytical method which does not suffer from manual line drawing.

The weakest interpretation in this manuscript in my opinion is around the trafficking in the TGN. It has become clear in recent years (The papers from the Bottanelli and Robinson labs), that the TGN is a complex organelle, which is effectively a post-Golgi network with a multimodal overlap to the endosomal system. There does appear to be a Golgi-proximal TGN but also a more distal TGN. I am concerned that the lack of subtlety in the final model and the authors view of the TGN risks oversimplifying the manuscript and undermining the resulting model. I encourage the authors to remove or revise this part of the model. As an additional point to the positive, the evidence that the TGN is the rate limiting step I consider well evidenced and important for inclusion and interpretation.

Another interpretation that I struggled to understand is the explanation of the monoexponential export of the TGN. In a way, this somewhat vindicates the Patterson interpretation, at least partially. The authors discuss peeling of cisterna, but if it is mono-exponential after a linear system there has to be a two phase, second system in the TGN. This is quite important because it is

the rate-limiting step the underlying biology here is the efficiency barrier to the secretory pathway.

I am usually pragmatic about the use of statistics and quantifications in manuscripts, and the authors have been very quantitative here. However, I cannot see (and I deeply apologise if I have missed this) what n refers to in the quantification. I think due to the controversy surrounding this topic it is very important that each experiment repeated multiple times independently. Especially as some of the data here is in direct contrast to other published peer-reviewed data.

A minor point is in the writing. On the whole the manuscript was a pleasure to read, however, one point I think has the potential to confuse readers. On line 57 the Patterson paper is cited after "especially as proposed by the cisternal maturation model" which makes it seem as if that paper proposed that model. I understand that the authors know this is not the case, so this should be easy to fix.

It is very hard on first read (even when looking at the text) to understand the difference between figure 1 C and D. It is clear in the legend; however I think this should be clarified in the figure as it is an important point and runs the risk of confusing the reader.

The words *cis*, *medial* and *trans* should be italicised throughout.

There are a number of missing figures (4C,D, 5C)- I have trusted the authors interpretations here, but would obviously need to see and evaluate these if invited for resubmission.

Referee #2:

This manuscript by Pirozzi et al. aims to address previously published findings that have challenged prevailing models of intra-Golgi transport, most notably the cisternal maturation model. Those previous findings were revisited with the aid of improved methodologies. The authors used electron microscopy, immunofluorescence microscopy, and live-cell fluorescence microscopy to analyze the intra-Golgi transit kinetics of protein cargoes and to map the site of cargo egress after intra-Golgi transit. Results from this study support important aspects of the prevailing models, including:

- a) After entering the Golgi, cargoes experience a delay or lag time as they transit the Golgi before they become incorporated into secretory carriers.
- b) Cargoes transit the Golgi in a cis-to-trans fashion with roughly linear kinetics, as predicted for a cisternal maturation mechanism.
- c) There is only limited exit of cargoes prior to reaching the trans-Golgi network (TGN).

Interestingly, the authors found that after intra-Golgi transit, cargoes accumulate in the TGN, and then they appear to exit the TGN with mono-exponential kinetics. This latter point helps to explain prior observations from Patterson et al. (2008) about Golgi export. The authors postulate that the TGN is a long-lived compartment that receives and accumulates secretory cargoes after they have moved through the Golgi stack via cisternal maturation, thus making the TGN a distinct entity.

Overall, the manuscript is clearly written and the experiments are logically executed and presented. The authors do a capable job of trying to reconcile their findings with those that challenged the maturation model, especially those from Patterson et al. The evidence for a lag time before cargo export from the Golgi is strong and is in agreement with observations by others. Moreover, the observed differences in lag times between cell lines is a valuable observation.

A concern with this work is the extensive use of nocodazole-induced mini-stacks to analyze intra-Golgi traffic. Although this approach has been used for more than two decades, mini-stacks do not necessarily reflect the behavior of the interconnected Golgi ribbon. While the overall arrangement of Golgi resident proteins is maintained in mini-stacks, significant structural changes take place in the absence of microtubules. Moreover, Golgi ribbon connectivity, which was used by Patterson et al. to rationalize their model, is absent in the mini-stack system. In addition, a careful analysis of Golgi mini-stacks has highlighted their structural and functional heterogeneity, a phenomenon that could significantly affect experimental outcomes (Fourriere et al., 2017; see Specific Critiques).

Specific Critiques:

1. The vesicle measurement and graph in Fig. 1E are confusing. Did the authors quantify vesicle signal outside the Golgi area or did they count individual spots/vesicles outside the Golgi area?
2. The intra-Golgi transit kinetics differ between experiments using the intact Golgi ribbon versus those with Golgi mini-stacks.

Intra-Golgi transit time appears slower in mini-stacks. The authors should comment on this difference.

3. The fractions of detected VSVG and hGH at the TGN are quite different in the immuno-EM experiments (Fig. 3 vs Fig. 4C). Is this difference due to the sensitivity of the antibodies? Or does it reflect the actual concentrations of the different cargoes at the TGN after intra-Golgi transit? If the latter is true, does this result imply that the two cargoes traverse the Golgi at different rates, or that they remain at the TGN for different lengths of time?

4. Based on the imaging experiments using intact ribbons (Fig. 1), the cargo transit time through the entire stack followed by incorporation into outgoing carriers is about 15 min. However, in immunofluorescence experiments using mini-stacks (Figs. 2 and 4), the bulk of the cargo has not reached the TGN at this time point. Instead, most of the cargo is just entering the trans-Golgi. How do the authors interpret this discrepancy? Is there a delay in intra-Golgi transport when nocodazole is used? Alternatively, might the trans-Golgi-to-TGN transport step the authors are proposing employ microtubules?

5. The authors state that the EM done by Patterson et al. did not include the TGN. However, the micrographs shown in that paper included additional area at both ends of the Golgi stacks. Therefore, the discrepancy between the current EM results and those from Patterson et al. should be more carefully discussed.

6. Unlike the other electron micrographs, Fig. 4C does not employ a label to indicate which side of the Golgi stack is CGN or TGN. Is there a specific reason?

7. For all of the mini-stack experiments, cells were exposed to nocodazole for only 3 hours. However, Fourriere et al. reported that during nocodazole treatment, the population of Golgi mini-stacks requires a longer time (around 6 hours) to mature into fully functional elements (J Cell Sci (2016) 129: 3238-3250). This longer incubation with nocodazole has been employed by Lu and colleagues in their analysis of intra-Golgi transport (J Cell Biol (2022) 221:e202109114). At shorter nocodazole treatment times, Golgi mini-stacks can receive cargo but may be defective in processing and export. This effect could represent a serious caveat for all experiments involving mini-stacks (see Experimental Suggestions).

8. On a similar note, how can the authors be sure that cargo accumulation in the TGN in Golgi mini-stacks is not an artifact caused by the absence of microtubules? Perhaps an intact microtubule network and the associated machinery are needed for efficient production of secretory carriers. This concern should be addressed.

9. The authors imply in lines 251-255 that the temperature-based protocol for a cargo wave works better than the RUSH system by allowing complete transfer of cargo from the ER to the Golgi. When RUSH is used, it seems that after 25 min there is still cargo influx from the ER, resulting in a broader distribution of cargo across the Golgi stack at later time points. However, it is unclear why the RUSH protocol would fail to generate a reasonably synchronized cargo wave. This point needs further clarification.

10. The graphs in Figs. 2B, 2C, 4B, and 5B show data points that fall outside the numerical limits of the y-axis. The numerical range of the y-axis should be expanded so that readers can associate the individual data points with actual numbers.

11. It is stated that Patterson et al. used COS7 cells for their experiments and that this cell line displays the shortest lag before cargo export, leading the earlier authors to think there was no lag between cargo entry into the Golgi and cargo export. However, Patterson et al. also performed iFRAP experiments with NRK cells and human fibroblasts, and the results were similar. This point should be addressed.

12. Spontaneous reversible photobleaching is known to occur for some fluorophores. Have the authors tested that their cargo constructs do not reversibly photobleach? Such an effect could influence the conclusions from the iFRAP experiment.

13. The authors should address the recent preprint by Lu and colleagues in which they show (using mini-stacks) that different cargoes display very different intra-Golgi transit times (<https://www.biorxiv.org/content/10.1101/2024.04.11.589010v1>).

Experimental Suggestions:

1. Repeat some of the key experiments after a 6-hour nocodazole treatment to confirm that the results are similar.

2. Measure cargo export from the TGN of Golgi mini-stacks in live cells. Exponential export kinetics from the TGN are only inferred to happen in mini-stacks based on the accumulation of cargo at the TGN at 60 min after cargo release from the ER. The authors should directly measure cargo egress from individual mini-stacks in live cells to strengthen this conclusion.

3. Perform electron microscopy of full Golgi ribbons as VSVG and hGH traffic through the Golgi, to compare the kinetics of transport in the intact Golgi versus mini-stacks.

Referee #3:

In this manuscript, Pirozzi et al. address the controversial issue of intra-Golgi transport and whether cargo actually traverses the stack from cis to trans. An alternative model states that cargo moves through the Golgi by rapid diffusion within a continuous cargo domain and exits without a delay. The new manuscript refutes the rapid diffusion model with new, improved data using multiple cell types. This paper essentially reaffirms what is shown in textbooks as the cisternal maturation model, and therefore addresses a fundamental aspect of biology. The manuscript also addresses the issue of whether cargo traverses the TGN before exit as opposed to exiting directly from the trans-Golgi. It finds that cargo does in fact reach the TGN and in fact briefly concentrates there on its way through the Golgi system.

The manuscript is well-written and the data is compelling and clear. The experiments use sophisticated iFRAP methodology and the latest technology for synchronized cargo transport to demonstrate that there is a clearly discernible lag between entry and exit from the Golgi and that during the lag, cargo can be visualized sequentially in the cis, medial, trans and then TGN compartments.

My only major concern with the data was in Figure 4, where the authors trace hGH transport. Immunofluorescence data in part A indicates that the bolus of synchronized hGH is located in the medial Golgi at 20 min post D/D solubilizer. However, immuno-EM in part C indicates that hGH is mostly in the TGN at that timepoint. While one may expect accumulation of hGH in the TGN at steady state, this is not a steady-state experiment. Thus, the data in panels A and C are in conflict, suggesting that an artifact may be present, which needs to be reconciled.

Another major concern was with the length and speculative nature of the Discussion, where the authors present a model for trans-Golgi-to-TGN transport. This model is not addressed or justified by the data in the paper. While the experiments demonstrate transport into and out of the TGN after a delay, they do not favor a particular mechanism for the process. In my opinion, the Discussion should be greatly shortened and the model removed and published in a different venue such as a review. It detracts from the elegance of the paper by introducing excessive speculation.

A minor point is that the authors use the term "temperature-independent transport" to describe their techniques which is confusing because obviously intra-Golgi transport is temperature-dependent. I suggest the term "temperature shift-independent transport".

Referee #4:

Summary:

In Pirozzi et al., the authors have addressed apparent contradictions / inconsistencies regarding the mode by which non-resident proteins are trafficked vectorially through the mammalian Golgi. The main focus of this investigation is to establish whether the vectorial transport model for intra-Golgi trafficking, whereby proteins are considered to transit the Golgi stack from cis to trans, is robust enough to challenge alternative models. The prevailing changing model to vectorial transport, proposed by Patterson et al., (2008), suggests that Golgi proteins do not undergo vectorial transport, but rather can exit from any cisterna in the apparent absence of any delay. The authors' findings, from a variety of cells types and using a number of methodologies, lead them to propose a novel transport model in which proteins are transported vectorially from cis to trans but thereafter accumulate in the TGN from where they are dispatched to elsewhere.

Critic:

This is a very comprehensive and carefully executed study. The manuscript is easy to follow and the figures are well presented and easy to navigate. The authors' conclusions are strongly supported by their data. In addition, the authors' presentation of the existing knowledge in the field of intra-Golgi transport is comprehensive and unbiased whilst at the same time clarifying and validating the importance of this study.

Concerns:

The manuscript would be improved by providing a more detailed description of methods used.

Figure 1:

For Figure 1E, I could not find a description of how the number of post-Golgi carries was determined. Similarly, it is not apparent from how many experiments these data were obtained, and what if any variability was observed. What is the relationship between "normalized" peak value and the number of carriers?

Figure 4:

As the immunoblot now appears it could be misunderstood that GAPDH is a loading control the secreted fraction (which it cannot be as the protein is not secreted). Perhaps a Ponceau S-stained membrane would suffice as assurance that more or less equal amounts of TCA precipitated proteins were loaded in each lane in the secreted portion of the IB?

Although the authors' mention predicted MW for the internal and secreted forms of hGH it would be helpful to future readers if these species were indicated in the immunoblot presented, particularly as more than one band is present, and mobilities with markers vary somewhat between panels.

In the secreted fraction it appears that uncleaved hGH is present. Can the authors explain this please? Similarly, it appears that processed hGH is present at t=0, how do the authors account for this.

Also "Time after DD-solubilizer addiction" "addiction" should presumably be "addition" instead.

Results:

Page 5: "The rapid partitioning model predicts that newly arrived cargoes exit the Golgi apparatus without any apparent delay, as would be expected if the cargoes have to traverse the Golgi apparatus before exiting it (Fig.S1)"

Is this what the authors intended to state?

We thank all the reviewers for their support and very constructive criticisms, which have immensely improved the quality of our manuscript. Below we provide a point-by-point response to the criticisms and explain the steps we have taken to address them.

Referee #1:

Pirozzi and Agliarulo et al. have used a combination of classical cell biology and modern cell biology techniques and approaches to revisit the well-worked problem of intra-Golgi transport. In particular they take advantage of modern methodologies and thinking to re-examine an influential manuscript from the Lippincott-Schwartz laboratory in which a bi-phasic model of Golgi sorting is proposed. This is a complicated and controversial field which is marked by serious disagreement. It is therefore important that any new contributions contribute seriously to our understanding and are backed by strong evidence, otherwise they run the risk of further complicating it. With that in mind, I think that this manuscript does indeed make a worthy contribution due to a set of well thought out experiments and conclusions. I have several points that I think should be addressed, and I think it is important (for the above reasons) that they are addressed properly.

We thank the reviewer for the appreciation of our work.

One of the issues in the manuscript to my mind is the switching from tsVSV-G to RUSH. It is well justified in the manuscript, but I think due to the absolute importance of comparable findings the experimental modality used in figure 1 should be used for the RUSH system.

We thank the reviewer for this suggestion, it is an important one. We have now repeated the iFRAP and the immunofluorescence-based experiments with a RUSH cargo (E-Cadherin-GFP) which showed similar results to that of temperature-sensitive VSVG-GFP. These results are now included in Figures 1C, D, and 2D. This further confirms that the observed delay in cargo exit in temperature shift experiments is not an artifact of the experimental protocol. Of note, while we also performed the same experiment with VSVG-GFP-RUSH for unknown reasons there was neither an efficient recovery of Golgi fluorescence after bleaching nor an efficient exit from the Golgi after iFRAP. So we used E-Cadherin-GFP-RUSH for our experiments.

In figures 2 and 4 the experimenters use a version of the GLIM methodology pioneered by the Lu lab. The authors are clearly aware of these contributions as they discuss them at length. There is a weakness in the analytical approach employed by this manuscript vs the work performed in the Lu lab, as the Lu approach models the Golgi ministack as a 2D gaussian rather than a line which results in higher quality centroid fitting. I encourage the authors to re-examine their data here and use this stronger analytical method which does not suffer from manual line drawing.

While the GLIM-based is likely to be statistically more sound due to the larger number of samples, the assay we performed had enough samples ($n=25$ in each of the three independent biological replicates) to be statistically optimal. In addition, as the reviewer has rightly pointed out there are analytical differences in the two methods. Nevertheless, our results are similar to the published results obtained by the GLIM method. Anyway, we have performed an experiment and quantified the results using the GLIM method as well as our linescan method which results in practically superimposing graphs. These results are now presented in Fig. S2C.

This results shows that our linescan method, while being simple still provides results that are dependable and reproducible.

The weakest interpretation in this manuscript in my opinion is around the trafficking in the TGN. It has become clear in recent years (The papers from the Bottanelli and Robinson labs), that the TGN is a complex organelle, which is effectively a post-Golgi network with a multimodal overlap to the endosomal system. There does appear to be a Golgi-proximal TGN but also a more distal TGN. I am concerned that the lack of subtlety in the final model and the authors view of the TGN risks oversimplifying the manuscript and undermining the resulting model. I encourage the authors to remove or revise this part of the model. As an additional point to the positive, the evidence that the TGN is the rate limiting step I consider well evidenced and important for inclusion and interpretation. We agree with the reviewer that we had probably oversimplified a bit while preparing the model and included concepts that were previously described in the field but not necessarily developed in this manuscript (like peeling off of cisterna, see below). So we have now simplified the model to restrict it to findings of this manuscript. The model now stands as follows:

Figure 6

The model remains faithful to the observations presented in the manuscript. 1. Cargo arrives at the cis-Golgi; 2. it is transported across the Golgi stack to the trans-Golgi. 3. From there it is shifted to TGN where it accumulates before exiting. During the transition from the trans-Golgi to TGN there is a change in the mode of transport which is not yet clear. From TGN the cargo then departs in carriers that are formed in a microtubule dependent manner (not shown) to post-Golgi compartments. TGN is shown as a compartment that is distinct from the Golgi stack and has exchanges with the endosomal compartment.

Another interpretation that I struggled to understand is the explanation of the monoexponential export of the TGN. In a way, this somewhat vindicates the Patterson interpretation, at least partially. The authors discuss peeling of cisterna, but if it is monoexponential after a linear system there has to be a two phase, second system in the TGN. This is quite important because it is the rate-limiting step the underlying biology here is the efficiency barrier to the secretory pathway

We do have a two-phase system of cargo transit across the Golgi apparatus. The linear part of transport across the Golgi stack (intra-Golgi transport) cannot be observed by the fluorescence technology (live imaging at low resolution) employed since the transport is from one part of the Golgi stack to another. This can be seen in the intra-Golgi transport assay, where the cargo movement across the Golgi stack is linear (Fig.2) and this is represented as the flat part in the iFRAP experiments (Fig.1C). The exponential part is the one that corresponds to the exit from the TGN (from perinuclear structure as seen in fluorescence imaging) and can be observed by the iFRAP methodology employed (see the monoexponential part of the graph in Fig.1C which is shown also in Fig.1D). We hope this explanation addresses the reviewers concern.

I am usually pragmatic about the use of statistics and quantifications in manuscripts, and the authors have been very quantitative here. However, I cannot see (and I deeply apologise if I have missed this) what n refers to in the quantification. I think due to the controversy surrounding this topic it is very important that each experiment repeated multiple times independently. Especially as some of the data here is in direct contrast to other published peer-reviewed data.

We showed representative experiments of biological replicates and n referred to the number of Golgi stacks (usually atleast 25 per time point) that we had analysed in that particular experiment. Now we have represented all three biological replicates together (indicated by differences in color: $n \approx 25 \times 3 \approx 75$ Golgi stacks in total). This is valid for Figs. 2B, 2C, 4B, and 5B. Further two independent researchers were involved in the analysis of replicates to maintain objectivity. We now mention this in the methods section.

A minor point is in the writing. On the whole the manuscript was a pleasure to read, however, one point I think has the potential to confuse readers. On line 57 the Patterson paper is cited after "especially as proposed by the cisternal maturation model" which makes it seem as if that paper proposed that model. I understand that the authors know this is not the case, so this should be easy to fix.

We have now changed the sentence to the following:

Patterson and colleagues, by studying the kinetics of cargo transport through and out of the Golgi based on different technical approaches, have challenged the concept of vectorial transport across the Golgi by cisternal maturation.

It is very hard on first read (even when looking at the text) to understand the difference between figure 1 C and D. It is clear in the legend; however I think this should be clarified in the figure as it is an important point and runs the risk of confusing the reader. We apologise for the confusion. We have now added a box in Fig.1C that denotes the area of the graph that is represented in Fig1D.

The words *cis*, *medial* and *trans* should be italicised throughout. We have done that.

There are a number of missing figures (4C,D, 5C)- I have trusted the authors interpretations here, but would obviously need to see and evaluate these if invited for resubmission.

This is not clear, we have all the figures. Indeed Referee #2 raises questions based on Fig. 4C.

Referee #2:

This manuscript by Pirozzi et al. aims to address previously published findings that have challenged prevailing models of intra-Golgi transport, most notably the cisternal maturation model. Those previous findings were revisited with the aid of improved methodologies. The authors used electron microscopy, immunofluorescence microscopy, and live-cell fluorescence microscopy to analyze the intra-Golgi transit kinetics of protein cargoes and to map the site of cargo egress after intra-Golgi transit. Results from this study support important aspects of the prevailing models, including:

- a) After entering the Golgi, cargoes experience a delay or lag time as they transit the Golgi before they become incorporated into secretory carriers.
- b) Cargoes transit the Golgi in a cis-to-trans fashion with roughly linear kinetics, as predicted for a cisternal maturation mechanism.
- c) There is only limited exit of cargoes prior to reaching the trans-Golgi network (TGN).

Interestingly, the authors found that after intra-Golgi transit, cargoes accumulate in the TGN, and then they appear to exit the TGN with mono-exponential kinetics. This latter point helps to explain prior observations from Patterson et al. (2008) about Golgi export. The authors postulate that the TGN is a long-lived compartment that receives and accumulates secretory cargoes after they have moved through the Golgi stack via cisternal maturation, thus making the TGN a distinct entity.

Overall, the manuscript is clearly written and the experiments are logically executed and presented. The authors do a capable job of trying to reconcile their findings with those that challenged the maturation model, especially those from Patterson et al. The evidence for a lag time before cargo export from the Golgi is strong and is in agreement with observations by others. Moreover, the observed differences in lag times between cell lines is a valuable observation.

We thank the reviewer for the positive comments.

A concern with this work is the extensive use of nocodazole-induced mini-stacks to analyze intra-Golgi traffic. Although this approach has been used for more than two decades, mini-stacks do not necessarily reflect the behavior of the interconnected Golgi ribbon. While the overall arrangement of Golgi resident proteins is maintained in mini-stacks, significant structural changes take place in the absence of microtubules. Moreover, Golgi ribbon connectivity, which was used by Patterson et al. to rationalize their model, is absent in the mini-stack system. In addition, a careful analysis of Golgi mini-stacks has highlighted their structural and functional heterogeneity, a phenomenon that could significantly affect experimental outcomes (Fourriere et al., 2017; see Specific Critiques).

We agree that there are significant structural changes between Golgi ribbon and ministacks in nocodazole treated cells. Nevertheless, several earlier studies from our own group and others have found ministacks to be a valuable and reliable model to study intra-Golgi transport. Further, we find nearly similar results from both ribbon and ministack Golgi (see below).

Specific Critiques:

1. The vesicle measurement and graph in Fig. 1E are confusing. Did the authors quantify vesicle signal outside the Golgi area or did they count individual spots/vesicles outside the Golgi area?

We apologise for the confusion. We counted individual spots/vesicles outside the Golgi area. We have now added the following sentence in the figure legend:

Carriers represent the fluorescence spots present outside the Golgi area.

2. The intra-Golgi transit kinetics differ between experiments using the intact Golgi ribbon versus those with Golgi mini-stacks. Intra-Golgi transit time appears slower in mini-stacks. The authors should comment on this difference.

In the live imaging experiments with intact Golgi ribbon, a small pulse of cargo was followed across the Golgi. While in ministack experiments the conditions were such that there is a continuous arrival of cargo which can “apparently” retard the movement of the peak across the Golgi. Nevertheless, there could still be a contribution of nocodazole-induced structural alterations in the Golgi that can potentially slow down the traffic as suggested by the reviewer. Since this was not the focus of the manuscript, we did not comment on it. But we agree with the reviewer this is important and points to an aspect of the Golgi that has been less explored. Given the importance we have now added a note about this in the revised manuscript, which reads as follows:

The cargo that was predominantly present in the cis-Golgi at 5 min after the release from the ER was found to be predominantly present in the trans-Golgi after 25 min of release from the ER (Fig.2A, B), which roughly corresponds to the 10-15 min intra-Golgi transport time we observed earlier in the HeLa cells (Fig.1B). Still there is an apparent delay of 5min in the intra-Golgi transport time of cargoes as measured by the immunofluorescence assay compared to iFRAP experiments (Fig.1C). Whether this is due to the absence of microtubules (following the use of nocodazole) in this assay needs further exploration.

3. The fractions of detected VSVG and hGH at the TGN are quite different in the immunofluorescence experiments (Fig. 3 vs Fig. 4C). Is this difference due to the sensitivity of the antibodies? Or does it reflect the actual concentrations of the different cargoes at the TGN after intra-Golgi transit? If the latter is true, does this result imply that the two cargoes traverse the Golgi at different rates, or that they remain at the TGN for different lengths of time?

In both cases we used the same anti-GFP antibody, so it is not due to a difference in the sensitivity of the antibodies and likely reflects their actual concentration. The observation of the reviewer is pertinent and we have now added the following note in the text:

Of note, the concentration of VSVG-GFP in the TGN is much less than what we had earlier observed with hGH-GFP suggesting that cargo-specific mechanisms of concentration are also at play here.

4. Based on the imaging experiments using intact ribbons (Fig. 1), the cargo transit time through the entire stack followed by incorporation into outgoing carriers is about 15 min. However, in immunofluorescence experiments using mini-stacks (Figs. 2 and 4), the bulk of the cargo has not reached the TGN at this time point. Instead, most of the cargo is just entering the trans-Golgi. How do the authors interpret this discrepancy? Is there a delay in intra-Golgi transport when nocodazole is used? Alternatively, might the trans-Golgi-

to-TGN transport step the authors are proposing employ microtubules? We think that there is no "discrepancy" between the experiments, given the differences in spatial and temporal resolution between the methods and also the differences in the mode of synchronisation. In the iFRAP method synchronisation is sharp since there is no more fluorescence cargo reaching the Golgi after 5min (we photobleach the ER). On the contrary, in the ministack experiments cargo continues to arrive from the ER once the block is released which tends to shift the peak more towards the cis-Golgi. In spite of these differences the "discrepancy" we find is roughly 5min which is beyond the temporal resolution of the ministack experiment. On the other hand, in iFRAP experiments spatial resolution is compromised to evaluate what fraction of the cargoes have reached TGN at 15min. Given these limitations we cannot emphasize on differences in kinetics between the experiments. To sum up while we do not disagree with the reviewer that nocodazole treatment resulting in the absence of microtubules may have affected one or more steps in the intra-Golgi transport or exit from the Golgi, we think these experiments are not enough to make that conclusion. Nevertheless the important point to note in these experiments is that transport from the cis to trans-Golgi happens and the exit from the Golgi happens at the TGN. As mentioned earlier we have now added a note about this in the manuscript.

5. The authors state that the EM done by Patterson et al. did not include the TGN. However, the micrographs shown in that paper included additional area at both ends of the Golgi stacks. Therefore, the discrepancy between the current EM results and those from Patterson et al. should be more carefully discussed.

Patterson et al counted the particles associated with the cisterna and not the tubular domain adjacent to the Golgi (see Fig.S2C in the Patterson et al 2008). We, on the contrary, have also included TGN in our calculations. So there is no discrepancy between our study and Patterson et al in terms of "concentration in the TGN" as measured by EM.

6. Unlike the other electron micrographs, Fig. 4C does not employ a label to indicate which side of the Golgi stack is CGN or TGN. Is there a specific reason?

No. We had done that particular EM experiment for other purposes and decided to use one image from that series to demonstrate the point of concentration in the TGN.

7. For all of the mini-stack experiments, cells were exposed to nocodazole for only 3 hours. However, Fourriere et al. reported that during nocodazole treatment, the population of Golgi mini-stacks requires a longer time (around 6 hours) to mature into fully functional elements (J Cell Sci (2016) 129: 3238-3250). This longer incubation with nocodazole has been employed by Lu and colleagues in their analysis of intra-Golgi transport (J Cell Biol (2022) 221:e202109114). At shorter nocodazole treatment times, Golgi mini-stacks can receive cargo but may be defective in processing and export. This effect could represent a serious caveat for all experiments involving mini-stacks (see Experimental Suggestions).

We and others have shown that 3h of nocodazole treatment at (37/40°C) is enough to allow the formation of functional Golgi stacks i.e. stacks that are polarized and transport cargo (Trucco et al 2004; Cole et al 1996). Fourriere et al had used a different protocol for complete depolymerization of MT (3h on ice with nocodazole followed by 30 min at 37°C). At the end of this incubation (effectively 30 min at 37°C) the ministacks that were forming were not "mature enough" to function. They then did a 6h and 24 h incubation

at 37°C (in the presence of nocodazole) after which they found the ministacks “mature” (monitored by the presence of Giantin). Thus, these experiments say that after 6h the ministacks are functional but do not say the minimum amount of time required for the functional maturation of the ministacks. Moreover, Lu and colleagues too use only 3h of nocodazole treatment in their studies as mentioned below in their study (Tie et al 2022):

“Intra-Golgi secretory transport of RUSH reporters

HeLa cells were transiently transfected to express GalT-mCherry and a RUSH reporter and were cultured in the presence of 16 nM His-tagged streptavidin. After 3 h of nocodazole treatment, 40 µM biotin and 10 µg/ml CHX were added to the medium, and cells were chased...”

Further, our earlier published study had also showed the presence of Giantin in the ministacks after 3h Nocodazole treatment suggesting these stacks were mature (Trucco et al 2004). Nevertheless, we have now performed Giantin staining and show that Giantin, GM130 and VSVG colocalize in nearly 100% of the ministacks supporting our conclusion that the ministacks that we use are mature and transporting (see below and new figure S3A,B).

8. On a similar note, how can the authors be sure that cargo accumulation in the TGN in Golgi mini-stacks is not an artifact caused by the absence of microtubules? Perhaps an intact microtubule network and the associated machinery are needed for efficient production of secretory carriers. This concern should be addressed. The EM experiments were done in the absence of nocodazole (Figs.3, 4C, 5C-D) and so this shows that cargoes accumulate in the TGN even when microtubules are present. We apologise that this was not clearly mentioned in the figure legend. we have now corrected this and add this note to the figure legend:

Cells treated as in Fig.2A (except for the nocodazole treatment) were fixed and prepared for EM.

9. The authors imply in lines 251-255 that the temperature-based protocol for a cargo wave works better than the RUSH system by allowing complete transfer of cargo from the ER to the Golgi. When RUSH is used, it seems that after 25 min there is still cargo influx from the ER, resulting in a broader distribution of cargo across the Golgi stack at later time points. However, it is unclear why the RUSH protocol would fail to generate a reasonably synchronized cargo wave. This point needs further clarification.

In temperature-based protocol, when cargo transport is studied (after ER exit) we usually shift the cells to 40°C (not performed in this manuscript) which prevents the further entry of cargo and a wave of cargo thus formed is followed (Trucco et al 2004). A similar effect is achieved by photobleaching in iFRAP experiments. With the RUSH system once the ER block is released it is not easy to put it back since biotin cannot be washed out. We have now added the following note in the manuscript:

This is likely due to the continued arrival of cargo into the Golgi (since once biotin is added it cannot be washed out due to high affinity binding with streptavidin), unlike with temperature-based protocols where this is apparently better controlled.

10. The graphs in Figs. 2B, 2C, 4B, and 5B show data points that fall outside the numerical limits of the y-axis. The numerical range of the y-axis should be expanded so

that readers can associate the individual data points with actual numbers. We apologise, we have now corrected them.

11. It is stated that Patterson et al. used COS7 cells for their experiments and that this cell line displays the shortest lag before cargo export, leading the earlier authors to think there was no lag between cargo entry into the Golgi and cargo export. However, Patterson et al. also performed iFRAP experiments with NRK cells and human fibroblasts, and the results were similar. This point should be addressed.

They performed only the TGN exit experiments in these cell lines. The 5 min cargo pulse experiment which allows to follow a pulse of cargo was done only in COS7 cells.

12. Spontaneous reversible photobleaching is known to occur for some fluorophores. Have the authors tested that their cargo constructs do not reversibly photobleach? Such an effect could influence the conclusions from the iFRAP experiment.

The total fluorescence did not significantly change during this period and indeed normalization to total fluorescence provided similar results (Fig. S2). So, if there was a reversible photobleaching it was minimal and did not affect the results.

Further if this process did occur, it will affect both ER and Golgi localized proteins and indeed would have led to an absence of the flat phase since Golgi localized “older” proteins will start leaving the Golgi without any lag, which we do not observe. So we conclude there was minimal, if any, reversible photobleaching.

13. The authors should address the recent preprint by Lu and colleagues in which they show (using mini-stacks) that different cargoes display very different intra-Golgi transit times (<https://www.biorxiv.org/content/10.1101/2024.04.11.589010v1>).

Thanks for pointing to this interesting manuscript. Differential transport kinetics of cargoes have been reported by us (Beznoussenko et al 2014) and others (Boncompain et al 2012, Fossati et al 2014). These could be because of faster intra-Golgi transport mediated by tubules (Beznoussenko et al 2014), differences in the rate of exit from the TGN and/or increased recycling of cargoes (Casler et al 2019, Fossati et al 2014). We have now added a discussion of this in the manuscript which reads as follows:

We propose that the TGN is a distinct and rate-limiting compartment before the exit of cargoes. This is in line with the earlier proposals of TGN as a distinct compartment in plant cells (Nakano, 2022; Tojima et al., 2019) and also explains the observed differential transport kinetics of cargoes out of the Golgi apparatus (Boncompain et al., 2012; Boncompain & Perez, 2013). But this does not explain other observations of differential intra-Golgi transport rate (Tie et al 2024). The latter could be because of faster intra-Golgi transport mediated by tubules (Beznoussenko et al 2014) and or increased recycling of cargoes (Casler et al 2019, Fossati et al 2014). These suggest there is still a lot more to understand in pathways of intra-Golgi transport.

Experimental Suggestions:

1. Repeat some of the key experiments after a 6-hour nocodazole treatment to confirm that the results are similar.

As mentioned earlier, we think 3h Golgi stacks are “mature” enough to transport. Further we have also checked the maturation status of the stack by the presence of Giantin as suggested by Fourriere et al. See new figure **S3**. We have now included the following note in the text:

It is important to note that a recent study has indicated that Golgi mini stacks in nocodazole-treated cells are functional after 6h of treatment and this can be ascertained by the presence of Giantin in the functional Golgi stacks (Fourier et al 2016). While this study does not say that the stacks are “non-functional” after 3h of treatment and we and others have used 3h of nocodazole treatment to restore functional ministacks (Cole et al 1996, Trucco et al 2004, Tie et al 2022). Nevertheless, we also ascertained in the conditions that we use most of the Golgi ministacks were functional by colocalizing Giantin, GM130 and VSVG in the stacks. As shown in Fig. S3A,B, almost all the nocodazole-induced ministacks were positive for all three markers suggesting that the Golgi stacks that we use are functional.

2. Measure cargo export from the TGN of Golgi mini-stacks in live cells. Exponential export kinetics from the TGN are only inferred to happen in mini-stacks based on the accumulation of cargo at the TGN at 60 min after cargo release from the ER. The authors should directly measure cargo egress from individual mini-stacks in live cells to strengthen this conclusion.

We have performed the experiment as suggested by the reviewer. We do observe a mono-exponential exit out of the Golgi ministacks. It is now incorporated in the text as Fig.S3C and we have now added the following text:

Accumulation of the cargoes in the TGN of the ministacks suggested that the cargoes will exit the Golgi in a monoexponential manner even from the ministacks (earlier we had observed a monoexponential exit from the Golgi ribbon, Fig1D). We tested this prediction with an iFRAP experiment and found that indeed the cargoes exited in a monoexponential manner (Fig.S3C) confirming their accumulation in the TGN compartment under this condition.

3. Perform electron microscopy of full Golgi ribbons as VSVG and hGH traffic through the Golgi, to compare the kinetics of transport in the intact Golgi versus mini-stacks. There was a mistake in the figure legend. The EM data in Fig.3 and other experiments were indeed done in the absence of nocodazole and with an intact Golgi ribbon. We have now corrected that and the legend reads as follows:

A. *Cells treated as in Fig.2A (except for the nocodazole treatment) were fixed and prepared by EM.*

This suggests that the directional transport from cis to TGN is not affected by the presence or absence of the microtubules.

While the study proposed by the reviewer is important in that it can reveal differences in the trafficking mechanisms of two very different cargoes and also highlight the contribution of Golgi organization to intra-Golgi transport, the result whatever it may be does not affect the conclusions of the present study that cargoes go from cis to trans Golgi and concentrate in the TGN before their eventual exit. So we would like to request to avoid this suggestion to compare VSVG and hGH traffic through ribbon and ministacks by EM which would be a lot of work and we will not be able to finish it within the stipulated time.

Referee #3:

In this manuscript, Pirozzi et al. address the controversial issue of intra-Golgi transport and whether cargo actually traverses the stack from cis to trans. An alternative model states that cargo moves through the Golgi by rapid diffusion within a continuous cargo

domain and exits without a delay. The new manuscript refutes the rapid diffusion model with new, improved data using multiple cell types. This paper essentially reaffirms what is shown in textbooks as the cisternal maturation model, and therefore addresses a fundamental aspect of biology. The manuscript also addresses the issue of whether cargo traverses the TGN before exit as opposed to exiting directly from the trans-Golgi. It finds that cargo does in fact reach the TGN and in fact briefly concentrates there on its way through the Golgi system.

The manuscript is well-written and the data is compelling and clear. The experiments use sophisticated iFRAP methodology and the latest technology for synchronized cargo transport to demonstrate that there is a clearly discernible lag between entry and exit from the Golgi and that during the lag, cargo can be visualized sequentially in the cis, medial, trans and then TGN compartments.

We thank the reviewer for the positive comments.

My only major concern with the data was in Figure 4, where the authors trace hGH transport. Immunofluorescence data in part A indicates that the bolus of synchronized hGH is located in the medial Golgi at 20 min post D/D solubilizer. However, immuno-EM in part C indicates that hGH is mostly in the TGN at that time point. While one may expect accumulation of hGH in the TGN at steady state, this is not a steady-state experiment. Thus, the data in panels A and C are in conflict, suggesting that an artifact may be present, which needs to be reconciled.

We have not done a quantitative analysis of intra-Golgi transport of hGH with all the required timepoints by EM. The figure in 4C was part of another series of unpublished study. It was chosen as it clearly demonstrates (by an alternate and high-resolution technology) that hGH (similar to other cargoes) does concentrate in the TGN. As the reviewer can see from data presented in Fig4B, even by 15min in some Golgi stacks hGH can be found in the TGN which further increases by 25min. So the image chosen was just fortuitous that showed TGN accumulation already by 20min. So there is no conflict between figures 4A-B and 4C. On the other hand, as the Referee #2 had pointed out there could be a difference in the rate of intra-Golgi transport between Golgi ministacks (IF experiments) and intact Golgi ribbon (EM experiments) which might explain the small difference of quantitative nature between Figures 4B and C. So we have now added two other images taken at the same time point to show the distribution of hGH in the Golgi stack (Fig. S4F,G).

Another major concern was with the length and speculative nature of the Discussion, where the authors present a model for trans-Golgi-to-TGN transport. This model is not addressed or justified by the data in the paper. While the experiments demonstrate transport into and out of the TGN after a delay, they do not favor a particular mechanism for the process. In my opinion, the Discussion should be greatly shortened and the model removed and published in a different venue such as a review. It detracts from the elegance of the paper by introducing excessive speculation.

We apologise. We have now simplified the model as mentioned earlier and have also shortened the discussion.

A minor point is that the authors use the term "temperature-independent transport" to describe their techniques which is confusing because obviously intra-Golgi transport is

temperature-dependent. I suggest the term "temperature shift-independent transport".
Thank you for the suggestion. We have done that now.

Referee #4:

Summary:

In Pirozzi et al., the authors have addressed apparent contradictions / inconsistencies regarding the mode by which non-resident proteins are trafficked vectorially through the mammalian Golgi. The main focus of this investigation is to establish whether the vectorial transport model for intra-Golgi trafficking, whereby proteins are considered to transit the Golgi stack from cis to trans, is robust enough to challenge alternative models. The prevailing changing model to vectorial transport, proposed by Patterson et al., (2008), suggests that Golgi proteins do not undergo vectorial transport, but rather can exit from any cisterna in the apparent absence of any delay. The authors' findings, from a variety of cells types and using a number of methodologies, lead them to propose a novel transport model in which proteins are transported vectorially from cis to trans but thereafter accumulate in the TGN from where they are dispatched to elsewhere.

Critic:

This is a very comprehensive and carefully executed study. The manuscript is easy to follow and the figures are well presented and easy to navigate. The authors' conclusions are strongly supported by their data. In addition, the authors' presentation of the existing knowledge in the field of intra-Golgi transport is comprehensive and unbiased whilst at the same time clarifying and validating the importance of this study.

We thank the reviewer for the enthusiastic support.

Concerns:

The manuscript would be improved by providing a more detailed description of methods used.

We made the methods section clearer and have also added a new section on Analysis of carriers (see below).

Figure

1:

For Figure 1E, I could not find a description of how the number of post-Golgi carriers was determined. Similarly, it is not apparent from how many experiments these data were obtained, and what if any variability was observed. What is the relationship between "normalized" peak value and the number of carriers?

We apologize for the confusion. The data presented refers to the Golgi apparatus from a single cell. Similar observations (formation of carriers after a delay) were observed in several experiments that were used in the study but since the magnification used for iFRAP experiments was low (20x) the carriers were not always clearly visible. So we did

two experiments with high resolution (63x) to monitor the cargo carriers. In both experiments, similar results were seen. We present one of them here. We have now added a note on how the quantitation was done in the methods section. The values presented both in the case of Golgi fluorescence and no. of carriers were normalized to the maximum value observed. This is presented as a normalized peak value.

Figure 4:

As the immunoblot now appears it could be misunderstood that GAPDH is a loading control the secreted fraction (which it cannot be as the protein is not secreted). Perhaps a Ponceau S-stained membrane would suffice as assurance that more or less equal amounts of TCA precipitated proteins were loaded in each lane in the secreted portion of the IB?

We have rearranged the figure so as to make it clear that GAPDH is not the loading control for the secreted fraction. The Ponceau S-stained membranes of the secreted fraction did not show any bands since the secretion study was done in the absence of serum. So unfortunately there is no loading control for secretion. Nevertheless what we want to demonstrate here is only of a qualitative nature that the secreted fraction of hGH is mostly furin cleaved and not of a quantitative nature, the latter would require a normalization as the reviewer has correctly pointed out.

Although the authors' mention predicted MW for the internal and secreted forms of hGH it would be helpful to future readers if these species were indicated in the immunoblot presented, particularly as more than one band is present, and mobilities with markers vary somewhat between panels.

We have now added markers to indicate the full-length and furin-cleaved forms of hGH. we have also added a note about them

The arrowhead indicates the full-length form of hGH-GFP-FM and the double arrowhead indicates furin cleaved form. The asterisk indicates a likely non-specifically cleaved protein since it is not secreted

In the secreted fraction it appears that uncleaved hGH is present. Can the authors explain this please? Similarly, it appears that processed hGH is present at t=0, how do the authors account for this.

We think the presence of an uncleaved fraction is likely due to the limiting nature of furin action. HeLa cells are not professional secretory cells made for secreting large amounts of cargoes and thus when a big pulse of cargoes is transported across the system may get saturated.

As for the presence of processed hGH already at t=0, we think that is not the furin processed form as the molecular weight is different. This is likely due to intracellular degradation of the overexpressed protein since this intermediate form is not secreted. Now we have added a note explaining this in the legend as mentioned earlier.

Also "Time after DD-solubilizer addiction" "addiction" should presumably be "addition" instead.

We apologize for the mistake. We have now corrected this.

Results:

Page 5: "The rapid partitioning model predicts that newly arrived cargoes exit the Golgi apparatus without any apparent delay, as would be expected if the cargoes have to traverse the Golgi apparatus before exiting it (Fig.S1)"

Is this what the authors intended to state?

Sorry, there was a mistake. We thank the reviewer for pointing it out. It should read as follows and we have not corrected it:

*The rapid partitioning model predicts that newly arrived cargoes exit the Golgi apparatus without any apparent delay, as would be expected if the cargoes **do not** have to traverse the Golgi apparatus before exiting it.*

Dear Dr. Parashuraman

Thank you for the submission of your revised manuscript to EMBO reports. We have now received the full set of referee reports that is copied below.

As you can see, the referees find that the study has been significantly improved during revision and recommend publication. Before I can accept the manuscript, I need you to address some minor points below:

- The manuscript sections should be in the following order: Title page - Abstract & Keywords - Introduction - Results - Discussion - Methods - Data Availability - Acknowledgments - Disclosure Statement & Competing Interests - References - Figure Legends - (Main Tables with legends if applicable) - Expanded View Figure Legends.
 - The conflict of interest statement should be removed from the Acknowledgments and provided in a separate "Disclosure and Competing Interests Statement" section.
 - The correct nomenclature for the movie in all places (source file name, title in the system, legend) should be Movie EV1; the legend should be removed from the manuscript and provided as a readme.txt file and then the movie and the legend should be zipped up together and uplidd as one zip folder called Movie EV1.
 - You have currently four Supplementary Figures. These could be converted to Expanded View figures. The nomenclature is Figure EV#, the legends are part of the main manuscript file, in a separate section called "Expanded View Figure Legends", which comes just after the main figure legends. The Supplementary text file you currently have is therefore obsolete. The methods need to be part of the main manuscript text and the figure legends would be in the manuscript as well. Please update the nomenclature in all places: source file names, titles in the manuscript tracking system, figure legends, and callouts in the manuscript.
 - We note that you currently list 5 corresponding authors. Our authorship policies conform to international standards for scholarly research journals in the biological sciences and we generally discourage listing of co-corresponding authors in excess of five. Could you please provide justification for the corresponding authorships? Thank you.
 - We further noted a few discrepancies between the names listed on the manuscript and in the online manuscript tracking system: Vincenzo Manuel Marzullo in the manuscript vs. Marzullo Vincenzo Manuel in the system; Alexander Mironov in the manuscript vs. Alexandre Mironov in the system. Please correct these.
 - The ORCID ID is still missing for co-corresponding author Alberto Luini. Please link the ORCID ID of Alberto Luini to his account in the manuscript tracking system.
 - Please provide a complete author checklist (reporting checklist for Life Science Articles), which you can download from our author guidelines (<<https://www.embopress.org/page/journal/14693178/authorguide>>). Please insert information in the checklist that is also reflected in the manuscript. The completed author checklist will also be part of the RPF.
 - Please add callouts for Figure 2D, Figure S4.
 - Please download and fill our Reagents and Tools Table template (.docx), which you can find in our author guidelines: <https://www.embopress.org/page/journal/14693178/authorguide#structuredmethods>. When submitting your revised manuscript, please do not include the Reagents and Tools Table in the Methods section of the manuscript but upload it as a separate file choosing the file type "Reagent Table".
 - The information in Appendix Table S1 and S2 can be transferred to the Reagents and Tools table.
 - Our production/data editors have asked you to clarify several points in the figure legends (see below). Please incorporate these changes in the manuscript and return the revised file with tracked changes with your final manuscript submission.
- A) Statistical test information. Only p-values that are actually shown in the figure panel(s) should (and must) be defined in the legends, all others should be removed from (or added to) the legend. Moreover, we ask for the specification of exact p-values:
- Please note that the exact p values are not provided in the legend of figure 2C
 - Please indicate the statistical test used for data analysis in the legend of figure S2 D
- B) Replicates and error bars:
- Please note that information related to n is missing in the legends of figures 3, S2 C.
 - Please note that the error bars are not defined in the legends of figures 3, S2 C.
- C) Data presentation:

- Please note that the red arrow heads are not defined in the legend of figure 3. This needs to be rectified.
- Please note that the blue circles are not defined in the legend of figure 3. This needs to be rectified.
- All quantifications need to show the individual data points in addition to the mean and error bars. E.g., Figure S3B: please add the individual datapoints to the means, i.e., the mean value from the 3 independent experiments. Similar for Figure 3.
- Rosaria Di Martino AC et al 2020 is a preprint. Please cite it as follows. In the text (preprint: Di Martino R et al, 2020). In the reference list as Author NAME1, Author NAME2 etc (YEAR) article title. bioRxiv doi [PREPRINT]
- As a standard procedure we edit the abstract of manuscripts to make them more accessible to all readers. Please find my suggested draft below my signature.
- Finally, EMBO Reports papers are accompanied online by
 - A) a short (1-2 sentences) summary of the findings and their significance,
 - B) 2-3 bullet points highlighting key results and
 - C) a schematic summary figure that provides a sketch of the major findings (not a data image).Please provide the summary figure as a separate file in PNG or JPG format at a size of 550x300-600 pixels (width x height). Please note that the size is rather small and that text needs to be readable at the final size. Please send us this information along with the revised manuscript.

With kind regards,

Martina

=====

Referee #1:

The authors have responded well to all of the comments and questions and addressed them appropriately. In my opinion, the manuscript should be published. There may be a reaction in the field as these debates continue, and I think this is a good thing. I look forward to seeing how these data and interpretations affect our understanding going forward.

Referee #3:

The authors have justified the apparent contradiction I pointed out between Figures 4A,B and 4C. I am no longer concerned about that point. They have also reduced the complexity of their model figure so as to not exceed the conclusions of this study, as well as shortened the Discussion section. I no longer have any major concerns that would prohibit publication in EMBO Reports.

Referee #4:

The authors have adequately addressed the comments and concerns I raised in my initial review.

=====

Abstract

The classical models of intra-Golgi transport envision a movement of cargoes from cis-to trans-Golgi followed by their sorting at the trans-Golgi network (TGN). During this vectorial transport, the cargoes are processed by sequentially acting glycosylation enzymes. A number of studies challenged the vectorial transport model and proposed alternative transport routes bypassing either directional transport or the TGN. We have re-visited intra-Golgi transport using varied cargo synchronization protocols,

employing both imaging and biochemical methods. We find that cargoes do move vectorially across the Golgi stack and reach the TGN. Cell type-dependent variations in transport kinetics and the limited resolution of fluorescence microscopy could have influenced earlier discrepant interpretations. Further, we find that exit from TGN is a rate-limiting step leading to the accumulation of cargoes there before their monoexponential exit. These findings support an intriguing model of intra-Golgi transport which involves classical vectorial transport across the Golgi stack followed by a non-maturation-based transport from the stack to the TGN and beyond.

All editorial and formatting issues were resolved by the authors.

Dr. Seetharaman Parashuraman
IEOS, CNR
Institute of Endocrinology and Experimental Oncology
Via P. Castellino 111
Napoli, NA 80131
Italy

Dear Raman,

I am very pleased to accept your manuscript for publication in the next available issue of EMBO reports. Thank you for your contribution to our journal.

Kind regards,

Martina
